# Single Image Test-Time Adaptation for Segmentation

**Klara Janouskova**  *klara.janouskova@fel.cvut.cz*
*Visual Recognition Group, Faculty of Electrical Engineering*
*Czech Technical University in Prague*

**Tamir Shor**  *tamir.shor@campus.technion.ac.il*
*Technion – Israel Institute of Technology, Haifa, Israel*

**Chaim Baskin**  *chaimbaskin@technion.ac.il*
*Technion – Israel Institute of Technology, Haifa, Israel*

**Jiri Matas**  *matas@fel.cvut.cz*
*Visual Recognition Group, Faculty of Electrical Engineering*
*Czech Technical University in Prague*

**Reviewed on OpenReview:** *https://openreview.net/forum?id=68LsWm2GuD*

## Abstract

Test-Time Adaptation (TTA) methods improve domain shift robustness of deep neural networks. We explore the adaptation of segmentation models to a single unlabelled image with no other data available at test time. This allows individual sample performance analysis while excluding orthogonal factors such as weight restart strategies. We propose two new segmentation TTA methods and compare them to established baselines and recent state-of-the-art. The methods are first validated on synthetic domain shifts and then tested on real-world datasets. The analysis highlights that simple modifications such as the choice of the loss function can greatly improve the performance of standard baselines. and that different methods and hyper-parameters are optimal for different kinds of domain shift, hindering the development of fully general methods applicable in situations where no prior knowledge about the domain shift is assumed.

Code and data: https://klarajanouskova.github.io/sitta-seg/

## 1 Introduction

A common challenge in machine learning stems from the disparity between source (training) and target (deployment) data domains. Models optimized to minimize an error on a dataset from a specific domain are often expected to perform reliably in different domains. The discrepancy between training and deployment data, known as the domain shift, is very common; in fact, few things do not change in time, and training happens (well) before deployment. A domain shift may substantially degrade model performance at deployment time despite proper validation on training data, yet it is often not explicitly addressed and most machine learning effort has focused on the generalization problem.

In many practical scenarios, the characteristics of the target domain are not known beforehand, making the preparation of the model with traditional domain adaptation Rodriguez & Mikolajczyk (2019); Tzeng et al. (2017) techniques nontrivial. Recent advances Wang et al. (2020b); Sun et al. (2020); Gandelsman et al. (2022) suggest that under certain weak assumptions about the domain shift - such as a stable label distribution across domains - it is possible to mitigate the performance degradation with methods based on the information carried by input data received at inference time.

Test-Time Adaptation (TTA) is suitable for a priori unknown or difficult-to-predict domain shifts. Characterized as an unsupervised and source-free technique, TTA adapts the model directly during inference. The

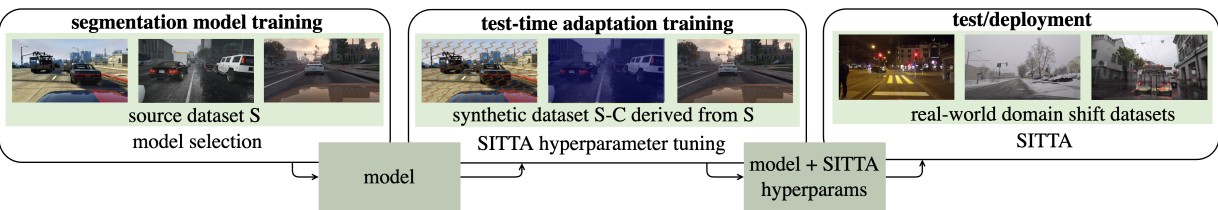

Figure 1: The proposed experimental framework for Single Image Test-Time Adaptation (SITTA). Hyperparameters are found on a synthetic dataset derived from the training set by applying a diverse set of corruptions. SITTA methods are then tested on real-world datasets with domain shift.

source-free nature, *i.e.* without access to the original training data, ensures compliance with data governance standards and enables adaptation in memory-constrained environments.

Single Image Test-Time Adaptation (SITTA) tailors a model at test time to each individual image. Since it operates on a single image, it does not introduce assumptions about the stability of the data distribution over time. Each time starting from the weights fixed at training time, SITTA is safe to use when any form of memorization of the deployment data is prohibited. A disadvantage is an increased computational time compared to batch methods and the lack of retaining the acquired knowledge. On the other hand, SITTA could be leveraged to reduce the computational cost of adapting to a sequence of similar images by only processing the most informative samples. Despite the advantages, only Khurana et al. (2021) primarily addresses SITTA, the mainstream of TTA research deals with continual test-time adaptation in a changing environment Niu et al. (2023); Volpi et al. (2022); Wang et al. (2022). These methods typically gradually update model parameters or accumulate image statistics for subsequent adaptation to individual images. Continual TTA strategies are practical in many applications, such as autonomous driving, but they are challenging from the point of view of accurate assessment of relative strengths and weaknesses. The difficulty arises from the evaluations conducted on many images with varying levels and kinds of domain shift. Moreover, the sequence in which images are presented can significantly influence performance metrics, adding a layer of complexity to assessing the true efficacy of these methods. SITTA streamlines the evaluation process and contributes to understanding the broader class of TTA strategies. Continual TTA analysis has focused on complementary issues such as catastrophic forgetting.

In this paper, we explore and improve the state of the art in Single Image Test-Time Adaptation for image semantic segmentation. While TTA has been applied to many tasks; the overwhelming majority of methods have been developed for image classification. Image segmentation TTA methods are each evaluated under very different conditions and compared to a limited set of baselines, making understanding their performance difficult. We focus on semantic segmentation SITTA with self-supervised loss functions. These methods are broadly applicable across various segmentation models and domains without being constrained by specific architectural or segmentation training requirements. Consequently, the many works that rely on batch normalization layers are not included in this work as they are incompatible[1] with the modern transformer architectures currently dominating the field. Likewise, methods such as image-reconstruction based TTA Wang et al.; Gandelsman et al. (2022) are not included due to their significant demands for training process modification and model architecture adjustments. Some methods require access to training data before deployment since they train an auxiliary network. However, they do not alter the training of the segmentation model.

Six TTA methods are evaluated in the SITTA setting. Two are established baselines: entropy minimization Hu et al. (2019) and pseudolabelling Lee et al. (2013) (self-training). They are the only ones evaluated in the SITTA setup before. The third method Prabhu et al. (2021) is a recently proposed segmentation TTA method that identifies confidently pseudolabelled pixels as consistent across augmentations. The fourth method Nguyen et al. (b) was proposed for image classification; we are the first to use it for image segmentation.

---

[1]At inference time, batch normalization uses per-feature mean and variance computed over the training dataset. TTA methods also incorporate the test data feature statistics. Transformer models typically use layer-normalization layers, which normalize features for each instance independently and thus always use the test sample statistics.

The main idea is that if a network is robust to small adversarial domain shifts (it is optimized to not change the prediction under an adversarial attack at test time), it will also be robust to real-world domain shifts.

The remaining two approaches are novel. We combine and extend two techniques optimizing self-supervised losses with 1. learnt segmentation mask domain discriminators Tzeng et al. (2017), so far used for unsupervised domain adaptation and never applied to TTA, and 2. Segmentation mask refinement [2] modules Karani et al. (2021); Valvano et al. (2021), never considered outside the medical domain TTA.

The idea of training-time unsupervised adaptation[3] with mask discriminators is that a classifier trained to distinguish masks from the source and target domains can be used to supervise adaptation. The source-domain segmentation model is finetuned to produce masks on the target domain that resemble those of the source domain, as predicted by the discriminator. This method is not directly applicable to TTA since the target images to train the discriminator are unavailable. However, the target domain masks can be synthesized. Copying random patches into different image locations was proposed in the medical domain. Such augmentation makes sense when the textures are more important than the shapes of objects, such as in the medical domain, but are not general enough for other images. We propose to use the output of the segmenter on adversarially attacked images. The adversarial optimization learning rate and the number of iterations control the severity of the domain shift. Further, Karani et al. (2021) proposes to replace the mask discriminator with a denoising autoencoder. Since a discriminator can use shortcuts and only focus on the most discriminative parts of the masks, a model that produces a refined mask is trained to consider all the masks. We explore both options.

We address the problem that arises from the practice in the current landscape of segmentation TTA that the performance assessment is carried out with inconsistent adaptation settings. For instance, keeping batch normalization statistics constant or updating them during entropy minimization can yield substantially different outcomes. If only one of the two options is tested Khurana et al. (2021); Volpi et al. (2022), contradictory results are reported. Often, the evaluation compares only with the established baselines such as entropy minimization Wang et al. (2020a) and batch normalization Ioffe & Szegedy (2015a) statistics adaptation Schneider et al. (2020); Nado et al. (2020), ignoring recent progress. Many methods that improve these baselines have been proposed in recent years for both segmentation and classification Nguyen et al. (a); Gao et al. (2023); Niu et al. (2023); Chen et al. (2022), and their relative merit is unknown, since a comprehensive comparison with a well-defined methodology is lacking.

Our experimental framework, depicted in Figure 1, consists of three stages: 1. **Segmentation model training**, which is standard. 2. **Test-time adaptation training** which tunes TTA hyper-parameters. We utilize an augmented version of the training datasets. This extension incorporates synthetic corruptions inspired by Hendrycks & Dietterich (2019). The corruption types include different kinds of noise and blur, weather conditions such as fog or frost, jpeg compression, and basic image intensity transformations. It can be derived from an arbitrary segmentation training dataset, as opposed to existing synthetic datasets used by previous work such as Ros et al. (2016). It provides precise control over the conditions and facilitates detailed analysis. 3. **Evaluation** on real-world test datasets with domain shift.

We experiment with two existing pretrained models for semantic segmentation. The models have different architectures and are trained on various datasets. Almost all segmentation TTA methods are evaluated on driving scenes benchmarks - we follow this practice and add a benchmark on common objects.

The main contributions of this paper are:

1. We conduct a comparative study of six TTA techniques run in SITTA mode for image segmentation: Two established baselines, two adapted state-of-the-art methods from image classification and continual TTA, and two proposed methods.

---

[2]Called "denoising autoencoders" in Karani et al. (2021).

[3]In the domain adaptation literature, the term "unsupervised" refers to the situation when images from the target domain are available to the method, but they are not labelled.In such a setting, training a source-target domain discriminator is possible, which contradicts the standard implication of the term "unsupervised". The setting where no samples from the target domain are available during training is referred to as "test-time adaptation" or "unsupervised source-free adaption".

2. Novel methods adapting ideas from unsupervised domain adaptation and medical imaging TTA to non-medical image segmentation are introduced, filling a gap in exploring diverse self-supervised loss functions. The method outperforms the other methods on multiple test datasets and is shown to be powerful on the images with the worst segmentation performance, as measured by Intersection over Union (IoU).

3. Improvements of baselines in the single-image setup by replacing Cross-Entropy (CE) with the IoU loss. The performance of pseudo-labelling is improved by 3.51 % and 3.28 % on GTA5-C and COCO-C validation sets while with CE loss, the improvements are by 1.7 % and 2.16 % only, respectively.

4. The first work shows the potential of SITTA for segmentation, an underexplored setup essential in applications with strict data governance standards or high variability among individual images.

## 2 Background

Common approaches to domain adaptation change the style of labelled source images to resemble the training images Tzeng et al. (2017); Zhang et al. (2018) or train domain classifiers to guide the adaptation process. In practice, this is not always feasible since source data may not be available for example for privacy or memory limitation reasons, or we may only have a small number of target domain images available when data arrive individually/in small batches, rather than all at once. In continually evolving environments, the distribution may change by the time adaptation on a large target dataset is completed. Various modifications of the traditional domain adaptation scenario tackling the aforementioned limitations have recently emerged, for example by considering no access to source data or a continual domain shift Liu et al. (2021a); Volpi et al. (2022); Wang et al. (2022); Bartler et al. (2022).

In particular, test-time adaptation methods assume no source data is available and aim to exploit the information from as little as a single target domain image. Like other domain adaptation methods, TTA methods are often inspired by semi-supervised learning methods. For instance, the most common TTA baseline relies on minimization of the predictions entropy, a method inspired by Saito et al. (2019). Other methods rely on adapting the batch normalization statistics, inspired by methods like adaptive batch normalization Li et al. (2018), or aggregating statistics to create so-called prototypes Tanwisuth et al. (2021) that can be used to build a classifier.

Some works also distinguish between TTA and Test-Time Training (TTT). The difference between TTA and TTT is that TTA methods such as Nguyen et al. (b); Karani et al. (2021) can be applied to arbitrary pre-trained models without any additional constraints while TTT methods like Gandelsman et al. (2022); Bartler et al. (2022); Liu et al. (2021b) require modifications to the training process. However, not all works make this distinction and the boundary is not always clear, as some methods like Karani et al. (2021) need to train an auxiliary deep net on the source data but do not modify the model pretrained weights. In this work, both will be jointly referred to as TTA for simplicity.

In Appendix A, other related domain adaptation scenarios and their relation to TTA are described.

Generally, TTA methods can be split into three groups: Adaptation in the input space, feature space and output space. **Input space adaptation** aims to translate the images from the source domain to the input domain. In practice, Gao et al. (2023) achieve this by feeding target images with added noise to a diffusion model trained on the source data, coupled with reconstruction guidance to preserve semantics. The model doesn't retain any knowledge from the adaptation - the advantage is it is not susceptible to catastrophic forgetting but the disadvantage is it may limit the adaptation capabilities. **Adaptation in the feature space** is the most common approach and typically relies on optimizing the network parameters via a self-supervised loss function. This can be done directly, i.e. through prediction entropy minimization, or by training an auxiliary task such as image reconstruction. Another set of feature-adaptation approaches are parameter-free and rely on accumulating the image statistics, such as the mean and variance of image features, or by aggregating confident prediction features into so-called prototypes, which are then used for classification. **Output space adaptation** techniques aim to improve the network output without neither altering the network parameters and statistics nor the input image. This is done for instance in Karani et al.

(2021) where an auxiliary network is trained to predict a refined mask. To the best of our knowledge, output space adaptation methods are typically only used to provide pseudo-masks, turning them into feature-space adaptation methods. This helps to iteratively improve the pseudo-masks and adapt to larger domain shifts. All the methods evaluated in this work can be considered as feature space adaptation methods, possibly via output space adaptation.

## 3 Related Work

**Test-Time Adaptation methods for classification.** Many recent methods propose improved strategies to update the batch normalization statistics Schneider et al. (2020); Nado et al. (2020). A limitation of these methods is the reliance on presence of batch nromalization, which is often not part of recent transformer-based architectures. In Wang et al. (2020b), the learnable parameters of the normalization layers are also updated via entropy minimization. While this method is often reported as unstable since single-image statistics may not be sufficient, the method can also only update the normalization layers learnable parameters, without the statistics update, making it generalizable to all currently used architectures.

On classification tasks, many methods outperforming the aforementioned baselines have been proposed. A combination of self-supervised contrastive learning to refine the features and online label refinement with a memory bank is proposed in Chen et al. (2022). Recently, a method based on updating the parameters of the normalization layers of the network by optimizing it for robustness against adversarial perturbation as a representative of domain shift was proposed in Nguyen et al. (b), outperforming similar test-time adaptation approaches. Rotation prediction is proposed in Sun et al. (2020) as self-supervised task to be learnt alongside the main one and then optimized at inference time. Lately, it was shown that reconstruction with masked auto-encoders is a very strong self-supervised task for test-time adaptation of classifiers by Gandelsman et al. (2022).

**Test-Time Adaptation methods for segmentation.** To the best of our knowledge, the only work also focused on adaptation to a single isolated image Khurana et al. (2021) is based on computing the statistics from augmented version of the input image, assuming batch normalization layers are present in the network. Both Prabhu et al. (2021) and Wang et al. (2022) exploit augmented views of the input images to identify reliable predictions. The method of Prabhu et al. (2021) is based on the consistency of predictions between augmented views, which replaces prediction confidence for selecting reliable pixels. Cross entropy loss is then minimized on such reliable predictions, together with a regularization based on information entropy Li et al. (2020) to prevent trivial solutions. The method achieves impressive results, however, in contrast to our experiments, knowledge of the target domain shift is used for hyper-parameter tuning. The evaluation assumed a full test set available at once, focusing on source-free domain adaptation, rather than TTA, but the method is applicable to the TTA setup as well. In Volpi et al. (2022), the performance of entropy minimization in a continual setup is explored, proposing parameter restart to tackle weight drift, significantly improving performance. The focus is on driving datasets only. Similarly, Wang et al. (2022) also focus on continual adaptation. Again, augmentations of the images are generated to obtain more reliable predictions. Further, the network parameters are stochastically reset to their initial values to prevent forgetting of the source domain knowledge.

**Test-Time Adaptation methods for medical imaging.** In Karani et al. (2021), an autoencoder is proposed that translates predicted masks into refined mask. At test time, the segmenter is optimized to produce masks closer to the enhanced ones. However, this work assumes the whole test dataset is available at once, in contrast to our single-image setup. The work of Valvano et al. (2021) is similar to Karani et al. (2021) but instead of a masked-autoencoder, a GAN-like discriminator trained end-to-end together with the segmenter is used, as well as an auxiliary reconstruction loss.

These works assume domain shifts specific to the medical imaging domain such as the use of a different scanner and thus make the assumption that only low-level features are affected. Under this assumption, these works typically optimize a small adapter only, ie. the first few convolutional layers of the segmenter. Nonetheless, these methods are generalizable to image segmentation.

**Enhancing existing TTA benchmarks.** There are multiple concurrent works that identify similar issues and reporting results consistent with our experiments, mostly for image classification. The work of Yu et al. (2023) also highlights the issue of evaluating each method under very different conditions and provides a benchmark for image classification TTA encompassing different adaptation scenarios, as well as diverse backbones and domain shift datasets. Similarly to ours, a significant disparity between synthetic corruptions performance and natural shifts is observed. However, the hyper-parameters were selected based on a single kind of domain shift, which may bias the results. Another work adressing the issue of fair comparison of TTA methods is that of Mounsaveng et al. (2024) which provides an analysis of existing orthogonal classification TTA methods. Class rebalancing is one of the tricks proposed to improve the methods' performance. Also, sample filtration to remove noisy high-entropy images is employed. In contrast, we analyze performance of different methods based on prediction entropy, showcasing some methods can actually be highly effective on those noisy, high-entropy samples. Similarly to ours, the work shows that baselines can be greatly improved by very simple tricks. In Niu et al. (2023), label imbalance at test-time is again identified as an important factor harming the TTA performance. Again, the works focus is on image classification and epxlores different normalization layer kinds and stabilization techniques of entropy minimization while we focus on comparions of cross-entropy and a class-imbalance aware segmentation loss function, the IoU. Finally, Yi et al. (2023) study TTA for image segmentations and how well classification methods transfer to semantic segmentation TTA. They conclude that many of the classification TTA improvements do not transfer to segmentation and again highlight the class imbalance, which is typically greater for segmentation datasets.

## 4  Methods

In total, six different methods are implemented and evaluated, including traditional TTA baselines, methods from other tasks and novel methods. All the methods consist of optimizing a self-supervised loss, the specifics of the loss being what differentiates the methods. It can be formalized as follows:

$$\theta_{i+1} = \arg\min_{\theta_i} \mathcal{L}(f_S^{\theta_i}, x)$$

where $x$ is the input image, $\theta_i$ are the parameters of the segmentation network $f_S^{\theta_i}$ at the $i$-th iteration and $\mathcal{L}$ is the self-supervised loss function.

The methods considered are:

- **Entropy-Minimization (Ent)**, a method proposed by Wang et al. (2020b) inspired by semi-supervised learning where the self-supervised objective is the prediction entropy. It has been used as a baseline by most of the TTA work. Only normalization layer parameters are typically updated to reduce the computational cost. Whether batch normalization statistics are updated or not varies.

- **Pseudo-Labelling (PL)**, also commonly referred to as self-training. The model is finetuned with pseudo-labels obtained from the pretrained segmentation model. There are many improvements and modifications. The standard approach is to threshold the predicted probabilities and only train the model on the most confident predictions.

- **Augmentation-Consistency (AugCo)**, proposed by Prabhu et al. (2021), is a method based on self-training enhanced by also optimizing for consistency between the original prediction and the prediction on augmented views, adapted to the single isolated image scenario.

- **Adversarial-Attack (Adv)** is the method proposed by Nguyen et al. (b) for image classification TTA, adapted to the single, isolated image segmentation.

- **Mask Refinement (Ref)** is one of the proposed methods and can be considered an enhanced pseudolabelling method. The pseudo-labels are obtained by a learnt refinement module that takes logit masks as inputs and outputs a refined segmentation mask. A method based on this idea has been implemented in medical imaging Karani et al. (2021) but never tested on non-medical tasks.

- **Deep-Intersection-over-Union (dIoU)**, the second proposed method is similar to Ref. However, a single-scalar quality estimate is predicted by a learnt module and minimized at test time. It is similar to using a GAN-like discriminator, as done in unsupervised domain adaptation literature Tzeng et al. (2017).

Only the necessary modifications to make the methods applicable to the single, isolated image segmentation setup with no assumptions of specific network architecture were applied to the existing methods.

The proposed methods based on learnt mask-refinement and mask-discriminator modules will be described in the rest of this section. A description of other methods and the details of their modifications in this work are in Appendix B.

The method proposed in this work is presented first and the differences from previous work are detailed aftwerwards.

**TTA with mask refinement** is based on the idea that since the output space changes much less than the input space, a mask translation module can be learnt to refine mask predictions on images from target distribution to resemble the masks obtained from source images. At test time, the refinement network can be viewed as an enhanced pseudo-label generation method. These pseudo-labels can then be used both as supervision for the segmenter or a direct replacement of the segmentation output without any parameter optimization. However, the second option is unlikely to tackle highly distorted masks since the refined mask cannot improve gradually.

To train the refinement network $f_{\mathrm{R}}^{\phi}$ with learnable parameters $\phi$, images from the source distribution and the pretrained segmentation network $f_{\mathrm{S}}^{\theta}$ are required. Given an image $x$ and $x'$ generated from $x$ by synthesizing a covariate domain shift (not changing the label), let us denote as $s = f_{\mathrm{S}}^{\theta}(x)$ and $s' = f_{\mathrm{S}}^{\theta}(x')$ the corresponding segmentation masks. Then, $f_{\mathrm{R}}$ is trained to predict $s$, given $s'$ as input:

$$\arg\min_{\phi} \mathcal{L}_{\mathrm{CE}}(f_{\mathrm{R}}^{\phi}(s'), s) \tag{1}$$

Predicted masks $s$ can also be replaced with ground truth $g$ at training time:

$$\arg\min_{\phi} \mathcal{L}_{\mathrm{CE}}(f_{\mathrm{R}}^{\phi}(s'), g) \tag{2}$$

where $\mathcal{L}_{\mathrm{CE}}$ is the cross-entropy loss.

At test-time, adapting to an image $x$, the model parameters are updated to minimize the IoU loss between mask prediction and a refined mask estimated by $f_{\mathrm{R}}$:

$$\theta_{i+1} = \arg\min_{\theta_i} \mathcal{L}_{\mathrm{IoU}}(f_{\mathrm{R}}(\overline{f}_{\mathrm{S}}^{\theta_i}(x)), f_{\mathrm{S}}^{\theta_i}(x)) \tag{3}$$

where $\theta_i$ are the learnable parameters of $f_{\mathrm{S}}^{\theta}$ at optimization iteration $i$ and $\overline{f}_S$ denotes no gradient flow throughout the computations of $f_{\mathrm{S}}$.

An overview of the training pipeline and the TTA with mask-refinement is in Figure 2.

**Refinement module training** requires generating masks resembling those that the model would output under domain shift. Since TTA assumes the domain shift is not known in advance, the goal is to synthetically generate diverse realistic segmentation masks representing masks predicted on images with domain shift. The advantage of the refinement module is that only the output space corrupted masks are needed. It doesn't matter how these were obtained since the refinement module is independent of the input images. This work simulates the mask corruptions by using mask predictions on the images from the first few iterations of a Projected Gradient Descent (PGD) Madry et al. (2017); Kurakin et al. (2018) adversarial attack, using the inverted mask as target. The more iterations of the attack, the higher the mask corruption, but the less realistic it becomes. Examples of generated corrupted masks are shown in Appendix C. The intuition

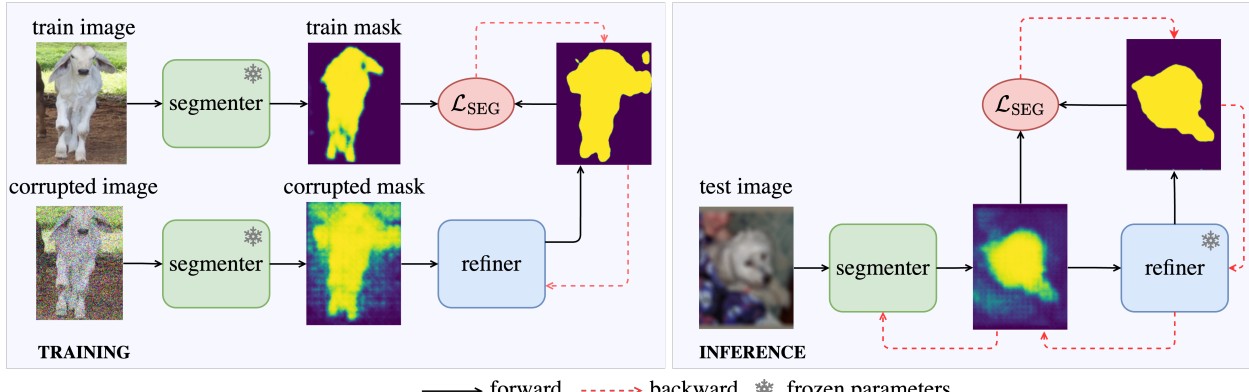

Figure 2: Mask refiner training (left) and Mask Refinement (Ref) TTA inference (right). During training, the segmenter outputs masks from a training image and a corrupted version of the training image simulating domain shift. The mask refiner is then trained to predict the clean image mask given the corrupted image mask as input only - no gradients flow back to the segmenter. At inference time, the segmenter output is fed into the refiner model. The refined output is then used as a pseudo-label to finetune the segmenter. A single gradient update is performed in each TTA iteration, then the masks are updated. The segmenter output may change with the updated weights, which in turn results in a new, possibly better, pseudo-label from the refiner. Visualized on single class prediction.

behind this adversarial approach is that in the first iterations, the most challenging pixels for the network are converted. Similarly, those image areas could be easily impacted by domain shift. The idea of using adversarial attack as a representative of domain shifts was also used in Nguyen et al. (b) and it was previously shown that adversarial robustness improves robustness to some domain shift kinds Croce et al. (2020).

**TTA with Deep-Intersection-over-Union** is almos equivalent to Mask Refinement (Ref). The only difference is that rather than learning a refined mask, the network is optimized to predict the value of the IoU loss. The IoU loss can be computed from ground-truth at training time but the network receives the predicted mask as input only.

**Difference from unsupervised domain adaptation literature:** There are many domain adaptation methods based on learning a discriminator between source and target domain. The key difference is that in this work, a module estimating how corrupted the mask is or a mask refinement module is learnt, instead of a binary classifier. Hence, the output is either a new segmentation mask, or a real number, rather than a class (domain) probability. This has the advantage that the mask needs to be considered as a whole, rather than allowing the network to focus on the most discriminative parts only. Also, the target domain is not known in advance and cannot be used to train the refinement network - masks resembling the target domain masks need to be synthesized.

**Difference from medical image segmentation TTA with denoising autoencoders** The idea of learning to predict a refined mask in the output space is the same as in in Karani et al. (2021). The main differences stem from the domain difference, since their approach is tailored to the medical domain. In Karani et al. (2021), the corrupted masks are obtained through swapping input image patches, a heuristic method with many hyper-parameters. Such augmentation makes sense when texture is more important than shapes, which is often not the case outside the medical domain. This work proposes exploiting adversarially-attacked images to synthesize corrupted masks. Further, this work excludes methods that make changes to the segmentation model architecture while Karani et al. (2021) rely on an additional normalization module. Their module architecture is designed with domain shifts such as intensity transformations, typical for the medical domain, in mind. Updates of parameters in the segmentation network itself are not allowed during TTA.

## 5  Experiments

The structure of this section is as follows:

1. **Evaluation metrics:** Given the focus of this study on SITTA and per-image performance analysis, we underscore the need for an image-level evaluation metric. The widely used mean Intersection over Union (mIoU) metric is typically applied at the dataset level and its adaptation for image-level assessment is not standardized.

2. **Experimental Setup:** Experiment settings shared across experiments such as network architectures or hyper-parameters. Creation of the synthetic SITTA training set derived from the segmentation training dataset is also explained.

3. **Experimental Results and Analysis:** Experiment results and analysis. The TTA methods are evaluated on two semantic segmentation models pretrained on the GTA5 Richter et al. (2016) and COCO Lin et al. (2014) datasets.

### 5.1  Evaluation metrics

The standard semantic segmentation evaluation metric is the mIoU, where the IoU score of each class is computed from predictions aggregated over the whole dataset and the per-class scores are averaged

$$\text{mIoU} = \frac{1}{\text{C}} \sum_{k=1}^{\text{C}} \text{IoU}_k(m_k, g_k) \tag{4}$$

where $m_k, g_k$ are the predictions and ground truth values for class $k$ for all pixels across all images. Concatenating all the masks into a single one and then computing the metric would not change the results, each pixel has the same weight independent of the image size or difficulty. This metric does not consider the size of objects or the difficulty of individual images. Per-image results cannot be compared, since not all classes are typically present in an image an it is not clear what value the score for that class should be.

Two additional metrics are introduced to account for the limitations of the standard mIoU and make the evaluation more fine-grained. The first metric is designed to consider class imbalance and difficulty of individual images, focusing on per-class performance. It will be referred to as $\text{m}\overline{\text{IoU}}_c$ and is defined as

$$\text{m}\overline{\text{IoU}}_c = \frac{1}{\text{C}} \sum_{k=1}^{\text{C}} \frac{1}{|\text{I}_k|} \sum_{i \in \text{I}_k} \text{IoU}(m_{ik}, g_{ik}) \tag{5}$$

where $\text{I}_k$ is the set of images in which either the prediction or the ground truth mask contains class $k$ and $|\text{I}_k|$ is the total number of images in $\text{I}_k$.

The second metric is focused more on per-image performance and can be computed for a single image. It will be referred to as $\text{m}\overline{\text{IoU}}_i$ and is defined as

$$\text{m}\overline{\text{IoU}}_i = \frac{1}{|\text{I}|} \sum_{n \in \text{I}} \frac{1}{|\text{C}_n|} \sum_{k \in \text{C}_n} \text{IoU}(m_{nk}, g_{nk}) \tag{6}$$

where $\text{C}_n$ is the set of classes in the predicted masks or the ground truth of image $n$. $I$ denotes the set of all images. This is the metric reported in our experiments unless stated otherwise. It allows for per-image performance comparison with the disadvantage of not accounting for class imbalance - less frequent classes (on the image level) get smaller weight.

Similar metrics were recently considered by other works Volpi et al. (2022), typically only aggregating over images where the given class appears in the ground truth (as opposed to either the ground truth or the prediction). This has the advantage that mistakes are only accounted for once, making the metric more

optimistic than ours. On the other hand, information about the errors is lost since the error is only computed for the ground truth class independently of what the incorrectly predicted class is.

On test datasets, the mDice (mean over the per-class dice scores Milletari et al. (2016)) and Accuracy (overall percentage of correctly classified pixels, regardless of class) metrics are also reported.

## 5.2 Experiment setting

**SITTA training set.** The SITTA training set for each model is derived from a set of 40 images from the segmentation model's training dataset extended with a set of 9 synthetic corruptions at three severity levels from Hendrycks & Dietterich (2019) such as blur, noise or fog, simulating different domain shifts. The original images are also included since the TTA methods should not harm the model on source domain images. Details about the corruption can be found in Appendix D. These synthetic datasets based on the GTA5 and COCO datasets are referred to as GTA5-C and COCO-C, respectively. Since the original images without corruption are also included, each SITTA training dataset consists of 1200 images (40 images, 9 + 1 corruption, three corruption levels).

**SITTA hyper-parameters.** For each TTA method, optimizing all the network parameters or normalization parameters is only considered, resulting in at least two different setups for each method. Further, when applicable (the methods compute a segmentation loss based on masks, as opposed to another self-supervised loss such as the prediction entropy), the CE and IoU losses are compared. While training segmentation models with a loss that takes class imbalance into account, such as the CE and the Dice loss Milletari et al. (2016), is standard, TTA work on image segmentation has relied on cross-entropy, which is also suboptimal from the point of view of the evaluation metric. It is desirable to align the optimization metric with the evaluation metric as much as possible. This results in four setups for the Ref, PL, and AugCo methods. The learning rate and number of TTA iterations are considered from learning hyper-parameters. The maximum possible number of iterations is 10 to limit the computational requirements. Reasonable learning rate values are found via a grid search and then extended with other promising values based on the initial results.

**Shared implementation details.** The refinement network architecture is a U-Net Ronneberger et al. (2015) with an EfficentNet-B0 Tan & Le (2019) backbone pre-trained on ImageNet from the Timm library Wightman (2019). It is trained with the AdamW Loshchilov & Hutter (2017) optimizer with a learning rate of $1e^{-3}$ and the Cross-Entropy (CE) loss. The SGD optimizer is used for the TTA since early experiments with AdamW showed a high divergence rate.

## 5.3 Experiment results

**GTA5 → Cityscapes, ACDC.** This experiment explores the performance of the TTA methods on a model trained on a synthetic driving dataset, GTA5, evaluating on real-world driving datasets under different weather conditions. The GTA5-pretrained model is the best-performing model of Volpi et al. (2022) (DeepLabV2).

| | Ent | | PL | | | | Ref | | | | AugCo | | | | Adv | | dIoU | |
|---|---|---|---|---|---|---|---|---|---|---|---|---|---|---|---|---|---|---|
| params | full | norm | full | full | norm | norm | full | full | norm | norm | full | full | norm | norm | full | norm | full | norm |
| loss | ent | ent | ce | iou | ce | iou | ce | iou | ce | iou | ce | iou | ce | iou | kl | kl | - | - |
| NA | 35.18 | 35.18 | 35.18 | 35.18 | 35.18 | 35.18 | 35.18 | 35.18 | 35.18 | 35.18 | 35.18 | 35.18 | 35.18 | 35.18 | 35.20 | 35.20 | 35.18 | 35.18 |
| $\text{TTA}_{\alpha^*}$ | 35.18 | 35.58 | 35.54 | 37.21 | 35.60 | 37.09 | 35.18 | **38.69** | 36.88 | 36.50 | 35.27 | 35.66 | 35.35 | 35.39 | 35.20 | 35.20 | 35.18 | 35.18 |
| $\Delta_{\text{ABS}}$ | $-\epsilon$ | 0.39 | 0.36 | 2.03 | 0.42 | 1.90 | $-\epsilon$ | 3.51 | 1.70 | 1.32 | 0.09 | 0.48 | 0.17 | 0.21 | $-\epsilon$ | $-\epsilon$ | $-\epsilon$ | $-\epsilon$ |

Table 1: $\text{m}\overline{\text{IoU}}_i$ results aggregated across corruptions and levels in the GTA5-C dataset, compared to non-adapted (NA) performance. The TTA hyper-parameters $\alpha^*$ were selected for overall best performance of each method. The **overall** and per-method best results are highlighted. No positive hyper-parameters are denoted by $-\epsilon$ (the performance converges to 0 from below).

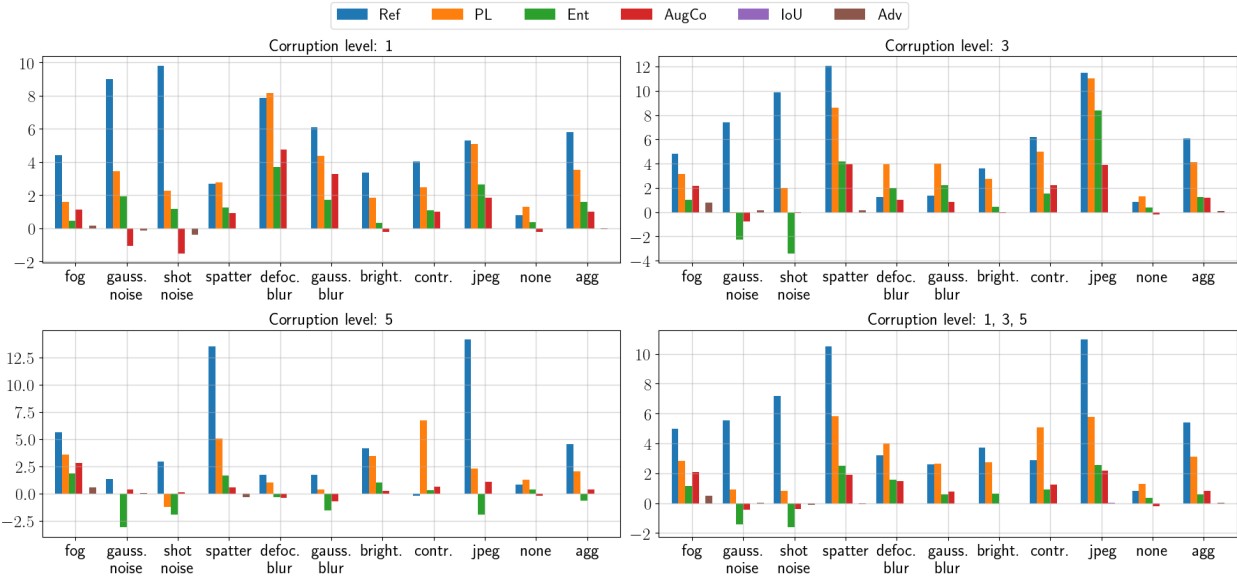

Figure 3: GTA5-C $\mathrm{m}\overline{\mathrm{IoU}}_i$ error reduction (%) depending on corruption levels. TTA with overall optimal hyper-parameters for GTA5-C.

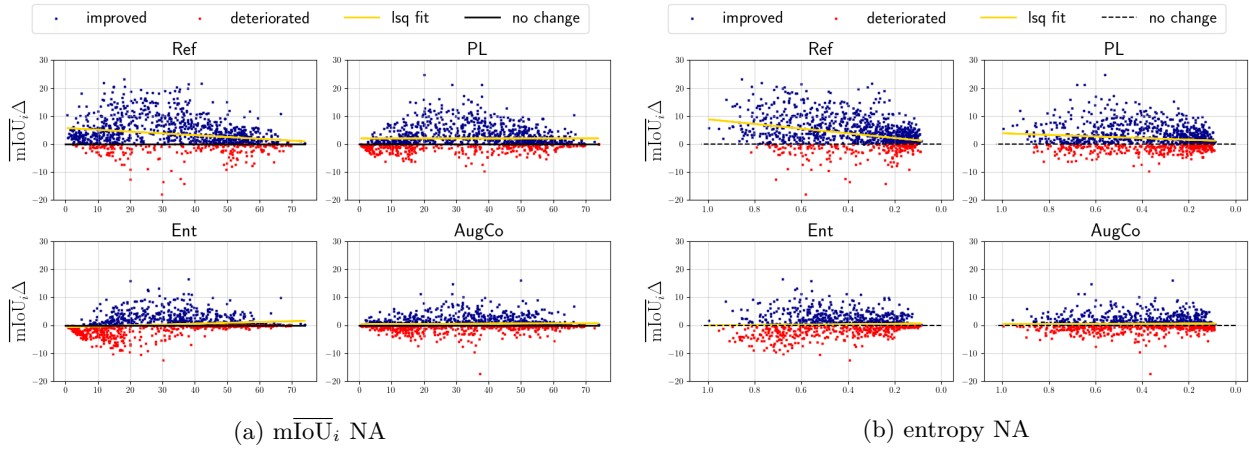

(a) $\mathrm{m}\overline{\mathrm{IoU}}_i$ NA

(b) entropy NA

Figure 4: The relationship between per-image scores (a) or entropy (b) before and the score after adaptation on the GTA5-C dataset. The difference between non-adapted (NA) $\mathrm{m}\overline{\mathrm{IoU}}_i$ or entropy and the $\mathrm{m}\overline{\mathrm{IoU}}_i$ after TTA is shown ($\mathrm{m}\overline{\mathrm{IoU}}_i\Delta$). A least-squares line fitted to the points is shown in yellow.

Since current methods do not consider different hyper-parameters for individual images, a single set with overall best performance across all corruptions and corruption levels is selected. The aggregated results with these overall optimal hyper-parameters on the SITTA training set can be found in Table 1. It can be observed that the biggest improvements are achieved either by PL with IoU loss, optimizing normalization parameters only, or by Ref with IoU loss, optimizing all the parameters. The best-performing method is Ref, improving by 3.51 % over the non-adapted baseline (NA). Other methods only marginally improve over NA or show no improvement at all. Optimizing CE generally yields worse results than optimizing the IoU. While updating normalization parameters only may stabilize Ent, optimizing all the parameters is essential for optimal performance of Ref. For other methods, the difference is smaller - optimizing normalization parameters only is faster and thus recommended.

In Figure 3, the total error reduction results with the same set of overall optimal hyper-parameters for each method are shown but for each corruption level and kind separately. It can be seen that it is not possible

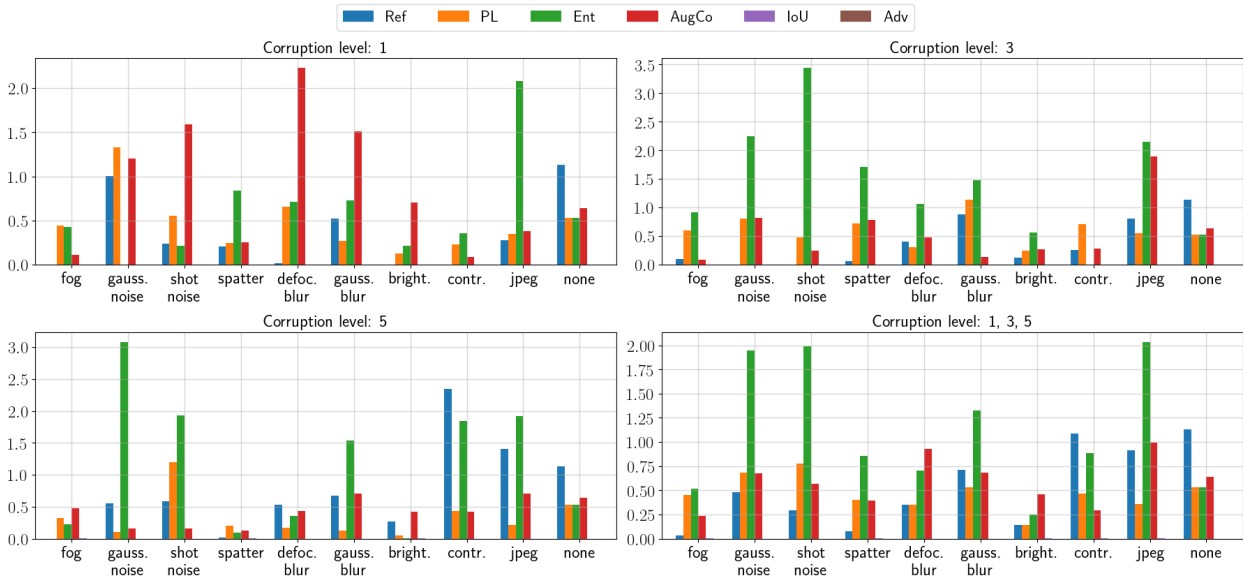

Figure 5: GTA5-C error reduction difference (%) between overall optimal hyperparameters and hyper-parameters selected for each corruption kind separately. The hyper-parameters were selected on GTA5-C.

to find a single set of hyper-parameters that would perform well across all the corruption levels with these methods. While all methods except for AugCo improve performance on level 1 corruptions, from level 3, negative results can be observed more often and many methods deteriorate/only slightly improve on level 5. Ref outperforms the other methods on the majority of corruption kinds and corruption levels. The aggregated results across all corruptions showed that the negative results are outweighed by the gains on level 1, resultin g in overall positive results for most of the methods.

In Figure 5, it is shown that if one could select optimal hyper-parameters for each corruption kind and level, results would improve substantially for many of the corruptions. This analysis suggests that unless the domain shift is known in advance, strategies with method and hyper-parameter selection for each image separately should be explored.

Only the methods with overall positive TTA results are considered for further analysis, namely Ref, PL, AugCo, and Ent. The relationship between the non-adapted (NA) performance and the performance improvement on individual images for different methods is visualized in Subfigure 4a. The analysis shows Ref outperforms other methods, especially on images that had low initial $\mathrm{m\overline{IoU}}_i$, while the performance of PL is more consistent across all initial scores but not as powerful for initial low scores. While Ent makes performance worse for low initial scores and improves more as the initial score increases, AugCo shows consistent improvements across all initial scores similarly to PL but to a smaller extent.

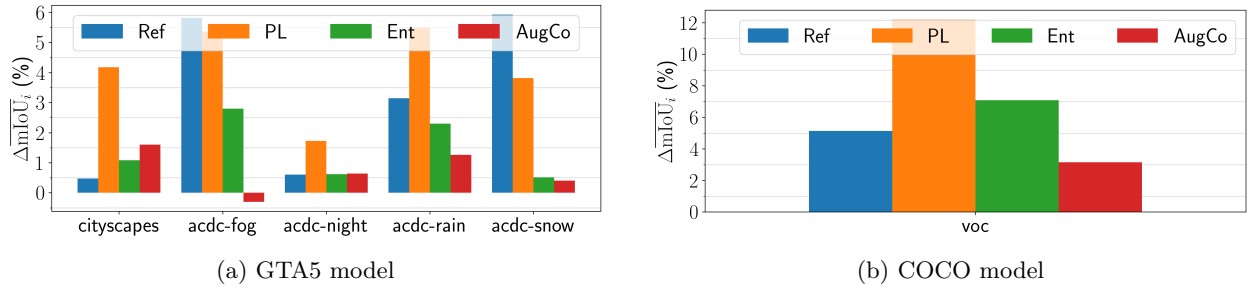

(a) GTA5 model

(b) COCO model

Figure 6: TTA $\mathrm{m\overline{IoU}}_i$ error reduction (%) on test datasets with hyper-parameters select on the TTA training datasets.

If the $m\overline{IoU}_i$ for each image were known (the ground truth is necessary for its computation), it could be used to either select a method performing best on those values or to select hyper-parameters. In Subfigure 4b, an analogous analysis is performed, replacing the $m\overline{IoU}_i$ with segmentation prediction entropy, which does not require any supervision. Similar results as with the $m\overline{IoU}_i$ can be observed.

After selecting the best hyper-parameters for each method on the SITTA training set, the methods are evaluated on 5 test datasets: ACDC-Rain, ACDC-Fog, ACDC-Night, ACDC-Snow, and Cityscapes. The Cityscapes represent a domain shift from synthetic to real images, while ACDC datasets add adverse weather conditions, making the domain shift even greater. The first four datasets are created by splitting the ACDC dataset by different conditions. Each of the test sets consists of 500 images. The test results are reported in Figure 6a and Table 2. Similarly to SITTA training datasets, either Ref or PL methods perform best, depending on the dataset. While not outperforming Ref on all the datasets, the performance of PL is consistently better than the other methods while Ref is outperformed or matched by the other methods on Cityscapes and ACDC-night. Table 2 shows that in standard metrics that do not consider per-image difficulty, AugCo outperforms the other methods in many cases.

| dataset | metric | method | | | | |
|---|---|---|---|---|---|---|
| | | NA | Ref | PL | Ent | AugCo |
| cityscapes | $m\overline{IoU}_i$ | 34.40 | 34.72 | **37.14** | 35.12 | 35.52 |
| | $m\overline{IoU}_c$ | 28.71 | 28.65 | **30.70** | 29.09 | 29.53 |
| | mIoU | 42.90 | 40.37 | 43.28 | 43.41 | **44.05** |
| | mDice | 55.78 | 52.53 | 56.04 | 56.17 | **56.89** |
| | Accuracy | 86.54 | 83.92 | 87.15 | 87.30 | **87.50** |
| acdc-fog | $m\overline{IoU}_i$ | 32.03 | **35.99** | 35.68 | 33.92 | 31.82 |
| | $m\overline{IoU}_c$ | 24.87 | 27.29 | **27.52** | 26.00 | 24.69 |
| | mIoU | 37.65 | 38.36 | **39.39** | 39.15 | 37.61 |
| | mDice | 51.42 | 51.12 | **53.29** | 52.68 | 51.51 |
| | Accuracy | 73.23 | **84.11** | 75.51 | 75.91 | 66.63 |
| acdc-night | $m\overline{IoU}_i$ | 13.60 | 14.12 | **15.09** | 14.13 | 14.15 |
| | $m\overline{IoU}_c$ | 10.77 | 10.96 | **11.53** | 10.68 | 11.01 |
| | mIoU | 15.79 | 13.77 | 15.11 | 15.53 | **16.25** |
| | mDice | 24.38 | 21.29 | 23.49 | 23.74 | **24.98** |
| | Accuracy | 52.09 | 52.47 | 51.86 | 52.82 | **53.09** |
| acdc-rain | $m\overline{IoU}_i$ | 33.52 | 35.60 | **37.16** | 35.05 | 34.50 |
| | $m\overline{IoU}_c$ | 26.15 | 27.40 | **28.47** | 26.89 | 26.73 |
| | mIoU | 36.93 | 36.21 | 37.61 | 37.53 | **37.98** |
| | mDice | 48.92 | 47.52 | 49.44 | 49.23 | **50.25** |
| | Accuracy | 84.56 | 84.22 | 85.65 | **86.13** | 84.74 |
| acdc-snow | $m\overline{IoU}_i$ | 31.54 | **35.60** | 34.15 | 31.87 | 31.81 |
| | $m\overline{IoU}_c$ | 25.28 | **28.09** | 27.16 | 25.38 | 25.45 |
| | mIoU | 35.30 | **37.89** | 36.64 | 35.34 | 35.40 |
| | mDice | 48.46 | **50.35** | 50.00 | 48.51 | 48.53 |
| | Accuracy | 73.17 | **81.52** | 74.48 | 73.66 | 73.45 |

Table 2: ACDC and Cityscapes test datasets results. Hyper-parameters were selected for overall best performance on GTA5-C. Best results for each dataset and metric are **highlighted**.

The inconsistencies of results between SITTA training and test suggest that unless the domain shift conditions are known in advance, it is difficult to select hyper-parameters based on a general SITTA training set.

**COCO → VOC**. In this experiment, the performance of TTA methods is studied on a model trained on the COCO dataset and evaluated on the VOC dataset. The segmentation model is an official Torchvision DeepLabV3 model with a Resnet50 backbone trained on the COCO dataset with a subset of 20 VOC classes.

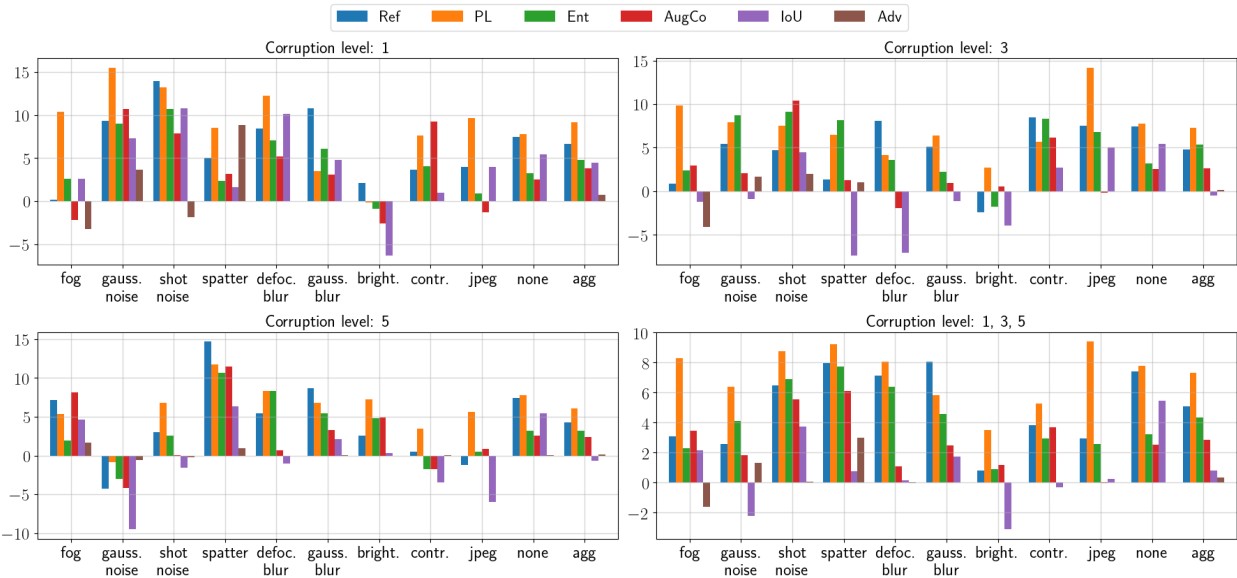

Figure 7: COCO-C m$\overline{\mathrm{IoU}}_i$ error reduction (%) depending on corruption levels. TTA with overall optimal hyper-parameters for COCO-C.

In contrast to previous experiments, it is a real-to-real dataset domain shift. The results of different methods with parameters selected for the overall best performance across all corruptions and levels can be found in Table 3. Biggest improvements are obtained by the PL and Ref methods. PL outperforms Ref, in contrast to the GTA5-C experiments. The best improvement is by 3.28 %, reducing the total segmentation error by 7.3 %. Consistently with previous experiments, best results are achieved with the IoU loss, outperforming CE in all cases. In contrast to GTA5-C, Ent achieves better results when optimizing all the network parameters, as opposed to optimizing only the normalization layer parameters. The same holds for PL. For Ref, optimizing all the parameters is again important. Other methods' improvements over the non-adapted baseline are marginal.

| | Ent | | PL | | | | Ref | | | | AugCo | | | | Adv | | dIoU | |
|---|---|---|---|---|---|---|---|---|---|---|---|---|---|---|---|---|---|---|
| params | full | norm | full | full | norm | norm | full | full | norm | norm | full | full | norm | norm | full | norm | full | norm |
| loss | ent | ent | ce | iou | ce | iou | ce | iou | ce | iou | ce | iou | ce | iou | kl | kl | - | - |
| NA | 55.01 | 55.01 | 55.01 | 55.01 | 55.01 | 55.01 | 55.01 | 55.01 | 55.01 | 55.01 | 55.01 | 55.01 | 55.01 | 55.01 | 55.16 | 55.16 | 55.01 | 55.01 |
| TTA$_{\theta*}$ | 56.97 | 56.75 | 57.17 | 57.99 | 57.10 | **58.30** | 56.24 | 57.31 | 56.56 | 57.16 | 55.40 | 55.59 | 55.30 | 56.30 | 55.16 | 55.16 | 55.61 | 55.74 |
| $\Delta_{\mathrm{ABS}}$ | 1.96 | 1.74 | 2.16 | 2.98 | 2.09 | 3.28 | 1.23 | 2.30 | 1.55 | 2.15 | 0.39 | 0.58 | 0.29 | 1.29 | $-\epsilon$ | $-\epsilon$ | 0.60 | 0.73 |

Table 3: m$\overline{\mathrm{IoU}}_i$ results aggregated across corruptions and levels in the COCO-C dataset, compared to non-adapted (NA) performance. The TTA hyper-parameters $\alpha^*$ were selected for overall best performance of each method. The **overall** and per-method best results are highlighted. No positive hyper-parameters are denoted by $-\epsilon$ (the performance converges to 0 from below).

The total error reduction results with a single set of optimal hyper-parameters for each method are reported for each corruption level and kind in Figure 7. The results slightly differ from those for the GTA5-C, as in this case, PL is consistently the best method, only rarely outperformed by Ref or AugCo. Negative results on some corruption kinds are already reported for level 1 corruptions.

In Figure 9, the results with optimal hyper-parameters for each method, corruption kind and level are shown. The results again improve substantially but this time, the differences between methods are smaller. Interestingly, the dIoU method performs much stronger than in the GTA5-C experiments.

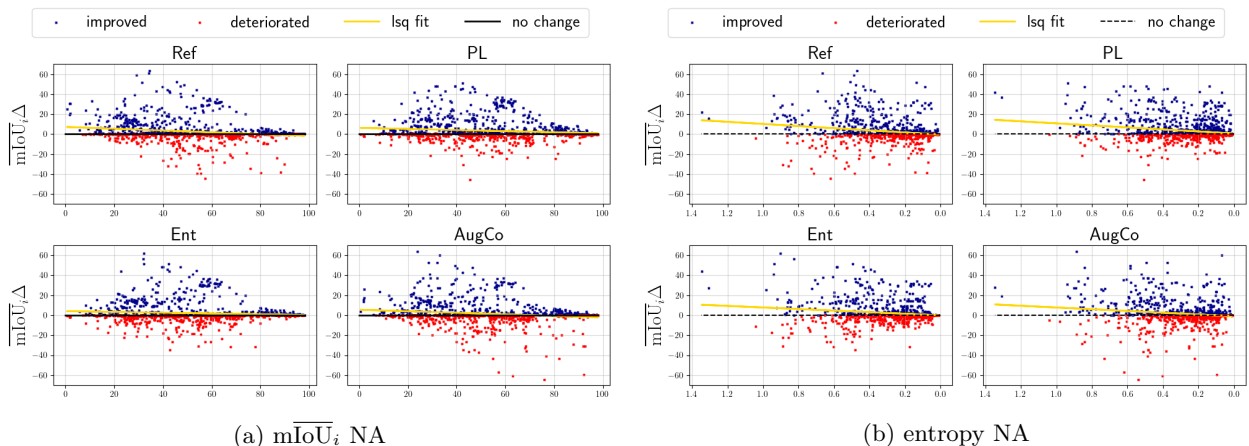

(a) $\overline{\text{mIoU}}_i$ NA

(b) entropy NA

Figure 8: The relationship between per-image scores (a) or entropy (b) before and the score after adaptation on the COCO-C dataset. The difference between non-adapted (NA) $\overline{\text{mIoU}}_i$ or entropy and the $\overline{\text{mIoU}}_i$ after TTA is shown ($\overline{\text{mIoU}}_i\Delta$). A least-squares line fitted to the points is shown in yellow.

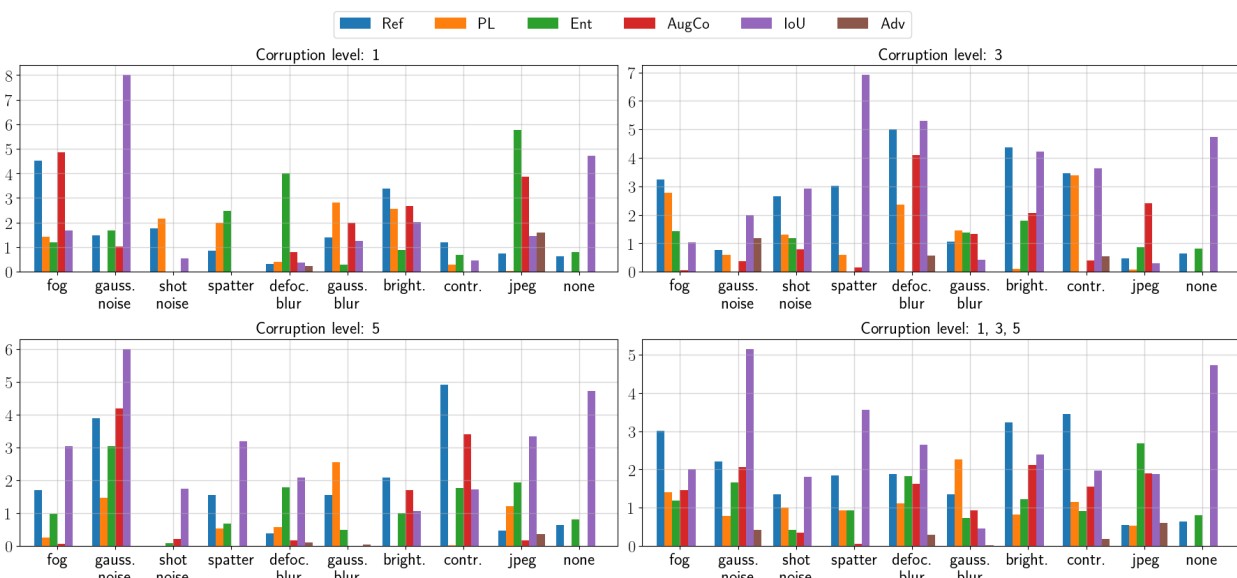

Figure 9: COCO-C error reduction difference (%) between overall optimal hyperparameters and hyperparameters selected for each corruption kind separately. The hyper-parameters were selected on the COCO-C.

Figure 10b compares the overall method performance on the SITTA training set. An oracle option is introduced where the method with the best results is picked for each image. There is a significant gap between the oracle and other methods, which further highlights that different methods are good in various cases and understanding the strengths of each methods can lead to significantly improved performance.

Again, only Ref, PL, AugCo, and Ent are used for further analysis. The relationship between the non-adapted (NA) performance and the performance improvement on individual images for different methods is visualized in Subfigure 8a. The distribution of initial non-adapted $\overline{\text{mIoU}}_i$ is different. The initial model is stronger than the GTA5 model. All methods show similar behavior - more improvement is achieved on images with a lower initial score.

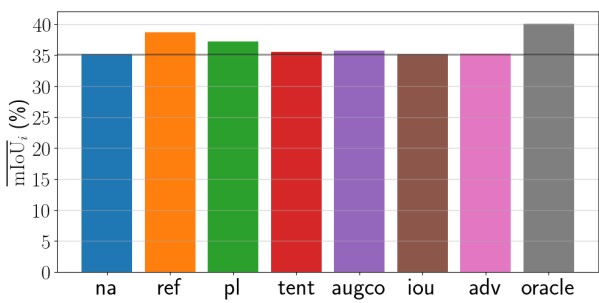

(a) Comparison of the overall performance of different methods on the GTA5-C validation set.

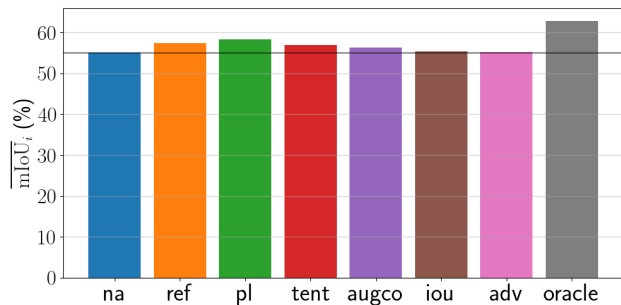

(b) Comparison of the overall performance of different methods on the COCO-C validation set.

Figure 10: The difference between non-adapted (NA) $\overline{\mathrm{mIoU}}_i$ .

The relationship of segmentation prediction entropy and $\overline{\mathrm{mIoU}}_i$ improvement by adaptation is shown in Subfigure 8b, supporting the notion that the entropy of prediction before adaptation is a good proxy for $\overline{\mathrm{mIoU}}_i$.

| | method | | | | |
|---|---|---|---|---|---|
| metric | NA | Ref | PL | Ent | AugCo |
| $m\overline{\mathrm{IoU}}_i$ | 77.00 | 77.62 | **79.67** | 78.85 | 77.75 |
| $m\overline{\mathrm{IoU}}_c$ | 63.38 | 63.38 | **66.90** | 65.38 | 63.10 |
| mIoU | 77.20 | **77.78** | 77.55 | 77.01 | 76.01 |
| mDice | 86.16 | **86.66** | 86.47 | 86.05 | 85.54 |
| Accuracy | 94.69 | **95.18** | 94.68 | 94.69 | 94.76 |

Table 4: VOC test dataset results. Hyper-parameters were selected for overall best performance on COCO-C. Best results for each dataset and metric are **highlighted.**

The results on the VOC test set are shown in Figure 6b and Table 4. PL outperforms other methods in the proposed metrics, all the methods improved over the non-adapted baseline. When considering standard metrics which do not consider per-image difficulty, Ref is best.

Additional results can be found in Appendix E.

## 6 Conclusions and limitations

This work investigated the performance of six Single Image Test-Time Adaptation (SITTA) methods on semantic segmentation problems. Two established baselines, two adapted state-of-the-art methods from image classification and continual TTA, and two novel methods were considered. We designed a framework that allows for hyper-parameter tuning and analysis of performance under diverse conditions on synthetic data inspired by Hendrycks & Dietterich (2019) that can be derived from an arbitrary training dataset. We evaluated the methods on real-world datasets.

Experiments on driving-scene datasets dominate the segmentation TTA literature. We followed this practice and experimented with models pretrained on the synthetic GTA5 dataset, evaluating on real-world driving scenes, including datasets with adverse weather conditions such as rain and fog. The common practice is to tune TTA hyperparameters on the Synthia Ros et al. (2016) datasets, however, this dataset contains the same weather and day-night conditions as the test datasets. Also, it is limited to driving datasets with the same set of classes. For this reason, we tuned hyper-parameters on a synthetic dataset derived from the training dataset by applying different corruptions, which gives us control over the domain shift conditions, facilitates analysis and is applicable to any existing dataset. We also added a novel benchmark where we evaluated a COCO-pretrained model on the VOC dataset, focusing on common objects.

We proposed two new methods inspired by ideas from unsupervised domain adaptation and medical imaging TTA. The stronger of the methods based on a learnt refinement module performed best on multiple of the test datasets and we showed that it is powerful on the images with the worst segmentation performance, as measured by IoU. We also showed that if we replace the IoU performance with the entropy of the predicted segmentation mask, which does not require the ground truth to be known, the same behaviour can be observed. This could be used to choose an appropriate method/hyper-parameters in future work.

We explored the effect of previously neglected design choices. Training with a loss function that accounts for class imbalance, a known issue of image segmentation datasets, such as the IoU or dice loss, is standard when training image segmentation models. When considering small batch sizes or even a single image, the class imbalance further increases, but SITTA methods implement baselines such as pseudo-labelling with the CE loss or entropy minimization with equal weight for all pixels. Our results revealed that while SITTA in the standard setting with CE loss did not improve performance much, substituting the CE with IoU improves performance substantially. The performance of pseudo-labelling was improved by 3.51 % and 3.28 % on GTA5-C and COCO-C validation sets while with CE loss, the improvements were by 1.7 % 2.16 % only, respectively. The experiments on whether to update all or normalization parameters only show this design choice significantly impacts results but the right option depends on the settings (method, dataset) - it is an important hyper-parameter to consider. Further, we find that entropy minimization, often reported as unstable for small batch sizes, performs well when the batch-normalization mean and variance are not updated at test time and only the affine parameters of the normalization layers are optimized.

In the GTA5-C synthetic datasets experiments, the refinement SITTA dominates, followed by the pseudo-labelling baseline. While the refinement is significantly better on some of the real-world test datasets, on other ones, pseudo-labelling performs best. In the COCO-C experiments, the top performers swap places: Pseudo-labelling is followed by refinement. On the test dataset, pseudo-labelling remains the best. While the other methods do not perform very well when TTA hyperparameters are optimized across many different corruption kinds, their performance improves when tuning them to specific kinds of domain shift. While there is not a single method performing best over all the test datasets, our results highlight the potential of SITTA for semantic segmentation.

**Limitations.** To limit the scope of the study, we only focused on adaptation with self-supervised loss functions and no reliance on batch normalization layers. While these methods tend to perform the best, their iterative optimization comes at an increased computational time. Methods alleviating this burden should be explored, such as only adapting to informative samples or methods inspired by efficient model finetuning.

The synthetic validation set created by applying artificial corruptions to the training set may not cover the complexity of real-world domain shifts. Label shift, common in real-world datasets, is not considered - the solutions are typically complementary.

Finally, only two models were considered and the effect of different model architectures on the individual methods is not known. While our work improves the understanding of TTA for semantic segmentation methods, a benchmark for fair and thorough evaluation of the methods is still missing.

### Acknowledgements

We greatly appreciate the help of Tomas Jenicek and Ondrej Tybl with proofreading of the manuscript.

Klara Janouskova was supported by the SGS23/173/OHK3/3T/13 research grant, and Jiri Matas was supported by the Technology Agency of the Czech Republic, project No. SS73020004 and project FACIS No. VJ02010041. Tamir Shor and Chaim Baskin were supported by NOFAR MAGNET number 78732 by the Israel Innovation Authority.

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
