## A    Related Domain Adaptation Scenarios

**Continuous domain adaptation** assumes the target data change continually, as opposed to a static target distribution. Furthermore, it is important to avoid catastrophic forgetting (previous knowledge). In this work, we make no assumption about the relationship between the distribution of subsequent samples, each sample could come from a different distribution. For this reason, our models are initialized to the pretrained weights before adapting to a new image and catastrophic forgetting is not a concern. While single-image adaptation methods can be extended to continual learning, it is not clear that methods performing better in the single-image setup will also perform better in batch or stream of data mode.

**Domain generalization** aims for a strong model that would generalize to unseen domains without any adaptation. Common approaches are domain-invariant representation learning, data augmentation or data generation. In contrast, this work focuses on adapting a pre-trained model to become a specialist in the current domain. It was shown in Volpi et al. (2022) that a stronger, more general model can lead to better adaptation results, making these directions complementary. However, without adaptation, training a very general model on a large set of distributions may harm the model's performance on the individual distributions when compared to specialized models with carefully optimized data augmentation Aquino et al. (2017); Steiner et al. (2021).

**Online domain adaptation** expects a stream of data as input, possibly a single one, as opposed to a whole dataset. Test-time adaptation methods can be sued for online adaptation but generally, online adaptation techniques do not assume the source data are not available.

**Zero-shot segmentation** requires the model to directly perform predictions for previously unseen classes without any adaptation to these classes.

## B    Baseline methods

In this section, the self-supervised loss functions optimized in the baseline methods are described, as well as the adaptations from the original implementation to the single-image setup when necessary.

### B.1    Entropy Minimization (Ent)

The Ent method minimizes the entropy of the segmentation predictions. In the context of learning with limited supervision, it was proposed in Grandvalet & Bengio (2004) for semi-supervised learning. In TTA, there is no labelled set that could be leveraged as regularization like in semi-supervised learning but the methods were shown to work for TTA as well Wang et al. (2020b). It was also shown that larger batch size and updating the parameters of the normalization-layers only improve stability of the method. But on segmentation, a dense-prediction task, adapting to a single image can lead to positive results, especially when not updating the batch normalization statistics but only the learnable parameters like in Volpi et al. (2022).

The method is simple, computationally efficient and widely adopted as a baseline. More formally, the method minimizes the entropy of the segmentation model predictions $s = f_S^\theta(x)$ for an image $x$:

$$\mathcal{L}_{\mathrm{Ent}} = \sum_{i=1}^{N} \sum_{i=1}^{C} \mathrm{s}_{ic} \cdot \log(\mathrm{s}_{ic}) \tag{7}$$

where C is the total number of classes, $\mathrm{s}_{ic}$ corresponds to the $i$-th pixel of the segmentation prediction s for class $c$ and N is the total number of pixels in the image.

In this work, the batch normalization Ioffe & Szegedy (2015b) statistics are not updated since it relies on the presence of batch normalization layers while many recent architectures use other normalization layers such as layer normalization Ba et al. (2016).

## B.2   Pseudolabelling (PL)

Pseudolabelling also comes from semi-supervised learning Lee et al. (2013) and is based on the idea of using the segmentation prediction of an image (prediction for each class binarized through argmax) as ground truth to optimize the model. In effect, it is the same as pseudo-labelling since both methods reduce class overlap. However, pseudolabelling has the advantage of allowing for different loss function. While the CE loss is typically optimized, our experiments show that in the single image setup, IoU leads to superior results.

## B.3   Adversarial Transformation Invariance (Adv)

This method is an extension of TIPI (Test-Time Adaptation with Transformation Invariance) by Nguyen et al. (b) to image segmentation. The main idea is to make the network invariant to adversarial transformation of the input image as a representative of domain shifts.

The optimization loss is computed as the reverse KL divergence loss between the model prediction $s' = f_S^\theta(x')$ where $x'$ is an adversarially transformed image and the prediction on the original input $s = f_S^\theta(x)$.

$$\mathcal{L}_{\text{Adv}} = \mathcal{L}_{\text{KL}}(s_i, s_i') \tag{8}$$

where $s_i'$ is the adversarially transformed prediction and KL is the Kullback-Leibler divergence loss defined as

$$\mathcal{L}_{\text{KL}}(p, q) = \frac{1}{N} \sum_{i=1}^{N} q_i \cdot \log(\frac{q_i}{p_i}) \tag{9}$$

In forward KL, p corresponds to the model prediction while q to the ground truth. Please note that, as suggested in Nguyen et al. (b), the reverse KL is used in the proposed method where the input arguments to the function are switched, compared to forward KL. Another important implementation detail is that the gradients should not flow through $s_i'$ - the tensor needs to be detached before the loss computation.

The same adversarial attacks in terms of the ground truth as for the IoU estimation and mask refinement methods are used to generate $x'$ but the computational complexity is reduced by using the Fast Gradient Sign Attack (FGSM) proposed in Goodfellow et al. (2014).

Importantly, two sets of batch normalization statistics are kept in Nguyen et al. (b), which is not done in our work due to the aim for general methods that do nto assuem the presence of specific network layers. This may be the reason why the methods perform poorly in our experiments. Another thing we ntoed is the high variance of the KL loss.

## B.4   Augmentation Consistency (AugCo)

The method of Prabhu et al. (2021) is adapted to the single image setup. The idea is to create two segmentation views based on the input image, both based on a random bounding box with parameters $\alpha$. The bounding box should take 25-50 % of the original image area and preserve its aspect ratio. View 1 is created by cropping and resizing the segmentation of the original image, $V_1 = \text{resize}(\text{crop}_\alpha(s), H, W)$ where $s = f_S^\theta(x)$ is the segmentation prediction for an image $x$ of spatial dimensions H, W. View 2 is created as the segmentation prediction on a cropped, resized and randomly augmented image, $V_2 = f_S^\theta(x')$ where $x' = \text{resize}(\text{crop}_\alpha(\text{jitter}(x)), H, W)$.

Finally, two masks are created to identify reliable predictions: Consistency mask based on consistency between the predictions of the two views, and confidence mask based on the confidence in the prediction in $V_2$, binarized with a confidence threshold $\theta$. These are then combined with the OR operation. We set $\theta = 0.8$.

The network parameters are then updated via pseudo-labelling based on predictions of $V_2$ and using reliable pixels only.

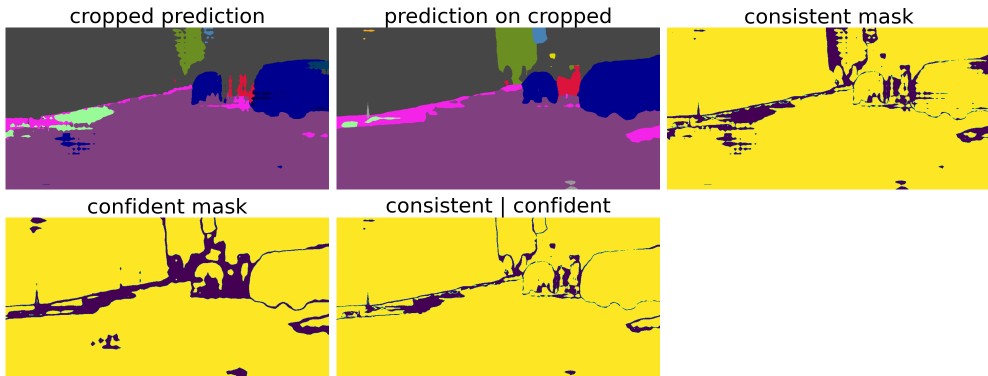

Figure 11: Visualization of the predicted views, confidence, consistency and reliablity (confident | reliable) masks used by the AugCo TTA method. Confident/consistent/reliable predictions are shown in yellow.

In Prabhu et al. (2021), an auxiliary information entropy loss preventing trivial solutions is also optimized. This loss requires running class-frequency statistics and is not applicable to the single-image setup. Further, an adaptive threshold based on per-class confidence distirbution in a batch of images is computed iinstead of a fixed threshold, which is also not applicable in our setup.

An example of the two views and the consistency and confidence masks, as well as the resulting reliability masks, are shown in Figure 11. For more details please see the original work.

## C   Adversarial refinement training

A visulaisation of mask evolution as the adversarial attack progresses is shown in Figure 12. It can be observed that the first iterations typically result in very small changes in easily confused areas, turning into more and more distorted masks.

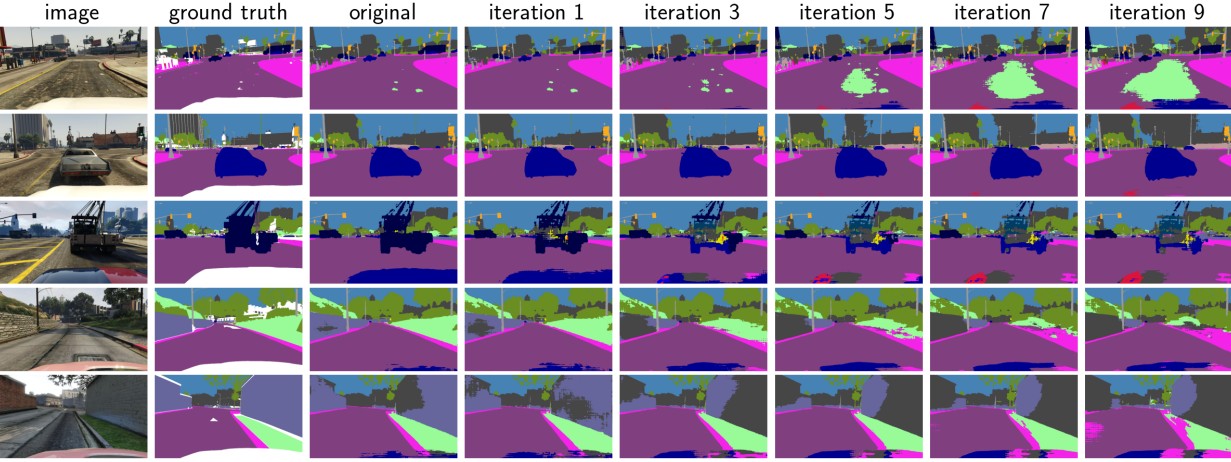

Figure 12: Evolution of masks over iterations of a projected gradient descent adversarial attack on the input image, the target being mask inversion for all of the classes. These masks serve as training data for the refinement module.

Instead of the iterative Projected Gradient Descent (PGD). However, since the projection only consists in restricting the output to a valid range for an image, typically implemented by simply clipping the output, it is also often referred to as iterative FGSM.

| corruption | description |
|---|---|
| **brightness** | is an additive intensity transformation, $x_c = \text{clip}(x+b)$ where $b$ controls the level. |
| **contrast** | is a multiplicative intensity transformation, $x_c = b(x - \overline{x}) + x$ where $b$ controls the level and $\overline{x}$ is a per-channel mean of the image intensities. |
| **frost** | first crops a portion of one of the frost image templates at a random location of the same size as the input image, $x_{\mathrm{f}}$. Then we compute $x_c = b_1 x + b_2 x_{\mathrm{f}}$ where the weights $b_1, b_2$ control the level. |
| **fog** | first generates a heightmap $x_{\mathrm{h}}$ using the diamond-square algorithm Fournier et al. (1982), where the wibble is controlled by a parameter $b_1$. It is then combined with the input image as $x_c = \frac{(x+b_1 \cdot x_{\mathrm{h}}) \cdot \max(x)}{\max(x)+b_1}$. |
| **gaussian noise** | is generated as $x_c = x + n$ where $n \sim \mathcal{N}(0, b)$ and $b$ controls the level. |
| **shot noise** | is generated as $x_c \sim \frac{\text{Pois}(x \cdot b, \lambda=1)}{b}$ where Pois denotes the Poisson distribution and $b$ controls the level. |
| **spatter** | simulates mud or water spoiling. The main idea of the algorithm is a combination of thresholding and blurring random noise. |
| **defocus blur** | first generates a disk kernel K with radius $b_1$ and alias blur $b_2$. The kernel is then used to filter each of the channels $x_c = \text{K}(x)$. |
| **gaussian blur** | corrupts the image by gaussian blurring $x_c = \mathcal{N}(x, b)$ where $b$ controls the level. |
| **jpeg** | is computed as $x_c = \text{jpeg}(x, b)$ where jpeg performs the JPEG compression with quality $b$. |

Table 5: Corruptions and their implementation details, a subset from Hendrycks & Dietterich (2019). The input of the transformation is an image $x$ normalized to the $(0, 1)$ range, the output is a corrupted image $x_c$. The clip function limits the values to the $[0, 1]$ range. This function is always applied to the output image after the transformation to obtain the final output $x_f = \text{clip}(x_c)$. For more details on the transformations and the values defining the level, please refer to the codebase.

## D Synthetic corruptions

The corruptions used in our experiments are a subset of the corruptions from Hendrycks & Dietterich (2019). An overview of the corruptions, as well as implementation details, can be found in Table 5

## E Additional experimental results

**SITTA training results**

The evolution of $\overline{\text{mIoU}}_i$ over TTA iterations depending on the hyper-parameters on the GTA5-C validation set can be found in Figure 14. The same results for the COCO-C validation dataset can be found in Figure 15.

**Test results** The $\overline{\text{mIoU}}_c$ comparison for all classes on the ACDC and Cityscapes test datasets can be found in Figure 13.

**Additional experiments** The refinement module is trained to predict a clean mask based on a corrupted mask simulating masks processed by the model under domain shift. The clean mask can be the segmentation prediction on clean, non-corrupted images or, when available, ground truth masks can be used instead. Comparison of these two choices is shown in Table 6. The results are somewhat inconclusive - for the COCO model it can be seen that the model trained on predictions is substantially better than the one trained on

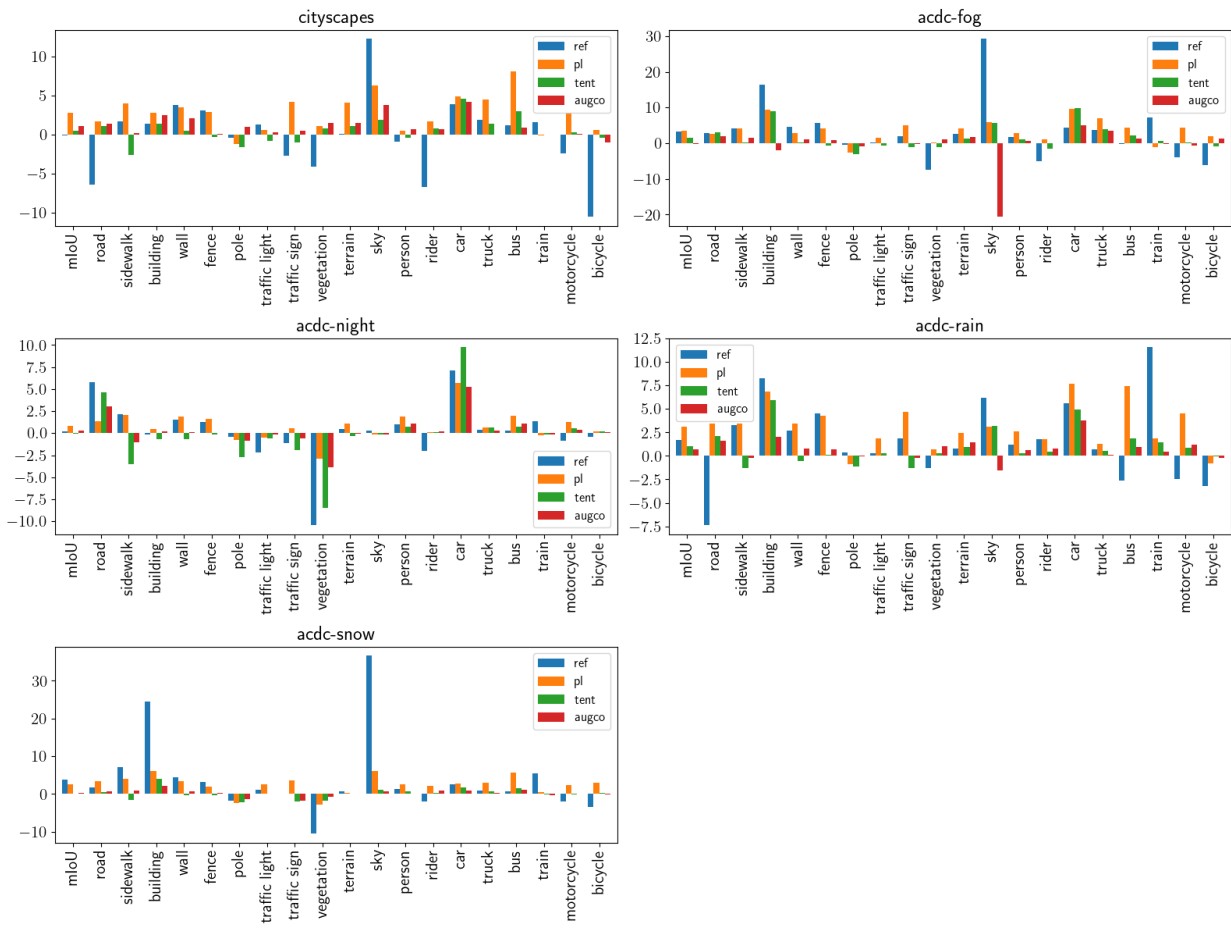

Figure 13: GTA5-C: $\overline{\text{IoU}}_c$ per-class comparison of different TTA methods.

|  |  | $\text{m}\overline{\text{IoU}}_i$ | |
|---|---|---|---|
| dataset | trained on | non-adapted | TTA |
| COCO-C | predictions | 55.01 | 57.31 |
|  | gts | 55.01 | 55.88 |
| GTA5-C | predictions | 35.18 | 38.63 |
|  | gts | 35.18 | 38.69 |

Table 6: Comparison of training the refinement module with ground truth masks and with segmentation model predictions. The $\text{m}\overline{\text{IoU}}_i$ aggregated across all corruption types and levels is reported with overall optimal hyper-parameters for each dataset.

ground truth. The GTA5 model performs similarly in both cases. One could argue that learning with GT can compensate for some of the mistakes even source distribution images, since prediction from output space back to output space is different then prediction from image to output space. On the other hand, when predictions differ significantly from ground truth even on source distirbution images, it can result in noiser data and more difficult training. The choice should be validated experimentally for each model and dataset.

**Visualization on test datasets** This part presents the visualizations of the Ref TTA method on the test datasets. The Ref method was selected because among the best performing methods, it is the most novel in the image segmentation setup and has shown particularly strong performance on images most severely

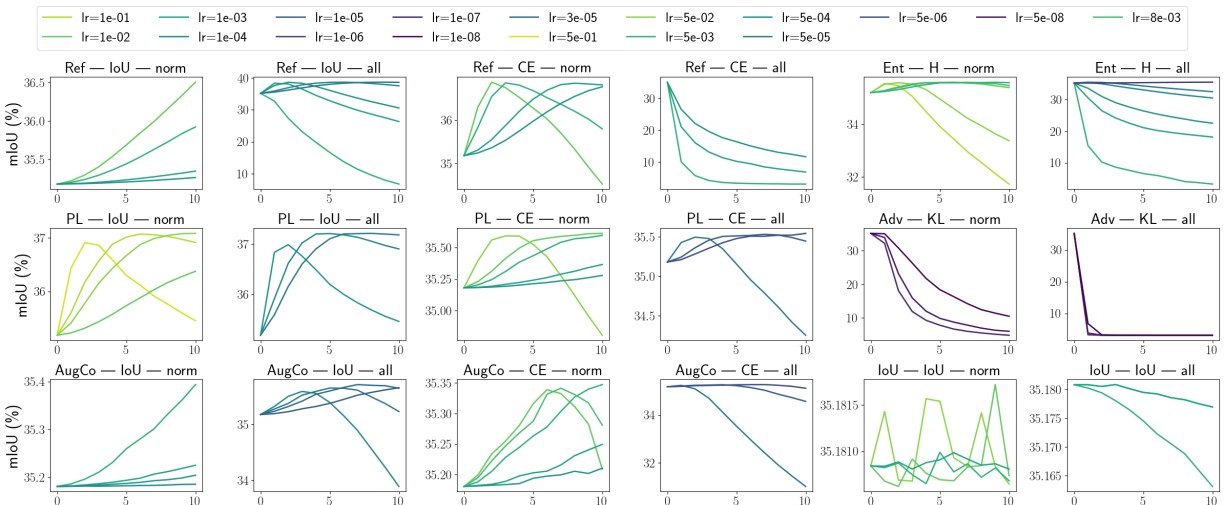

Figure 14: GTA5: $m_i\overline{IoU}$ evolution over 10 TTA iterations as a function of the learning rate. The results are reported as 'method - loss - optimized parameters'. The y-axes scale differs for each subplot to better visualize learning-rate differences for each method.

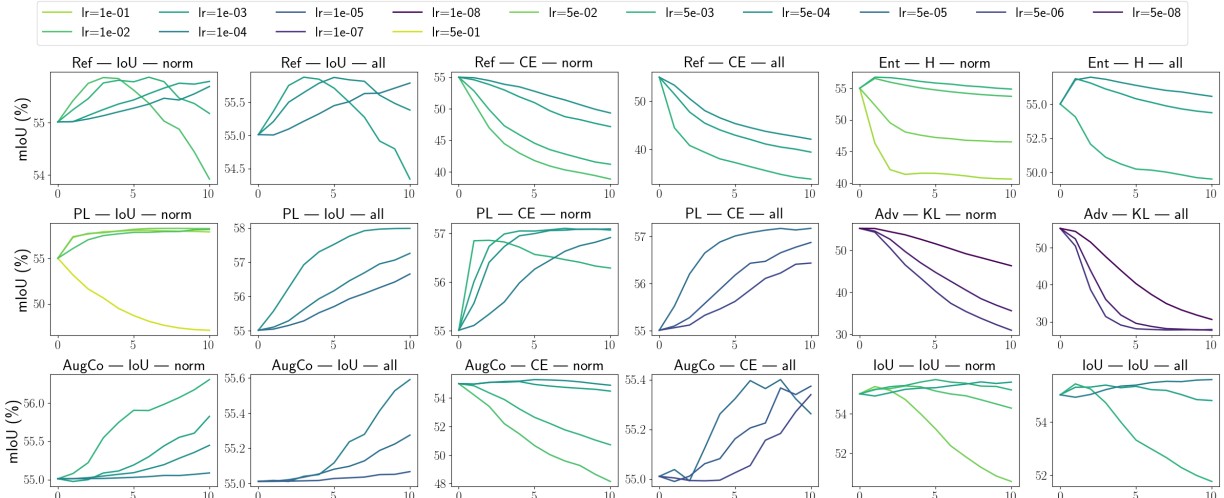

Figure 15: COCO-C: $m_i\overline{IoU}$ evolution over 10 TTA iterations as a function of the learning rate. The results are reported as 'method - loss - optimized parameters'. The y-axes scale differs for each subplot to better visualize learning-rate differences for each method.

impacted by domain shift. The visualizations can be found in Figure 16 (VOC), Figure 17 (ACDC-fog), Figure 18 (ACDC-night), Figure 19 (ACDC-snow), Figure 20 (ACDC-rain) and Figure 21 (Cityscapes).

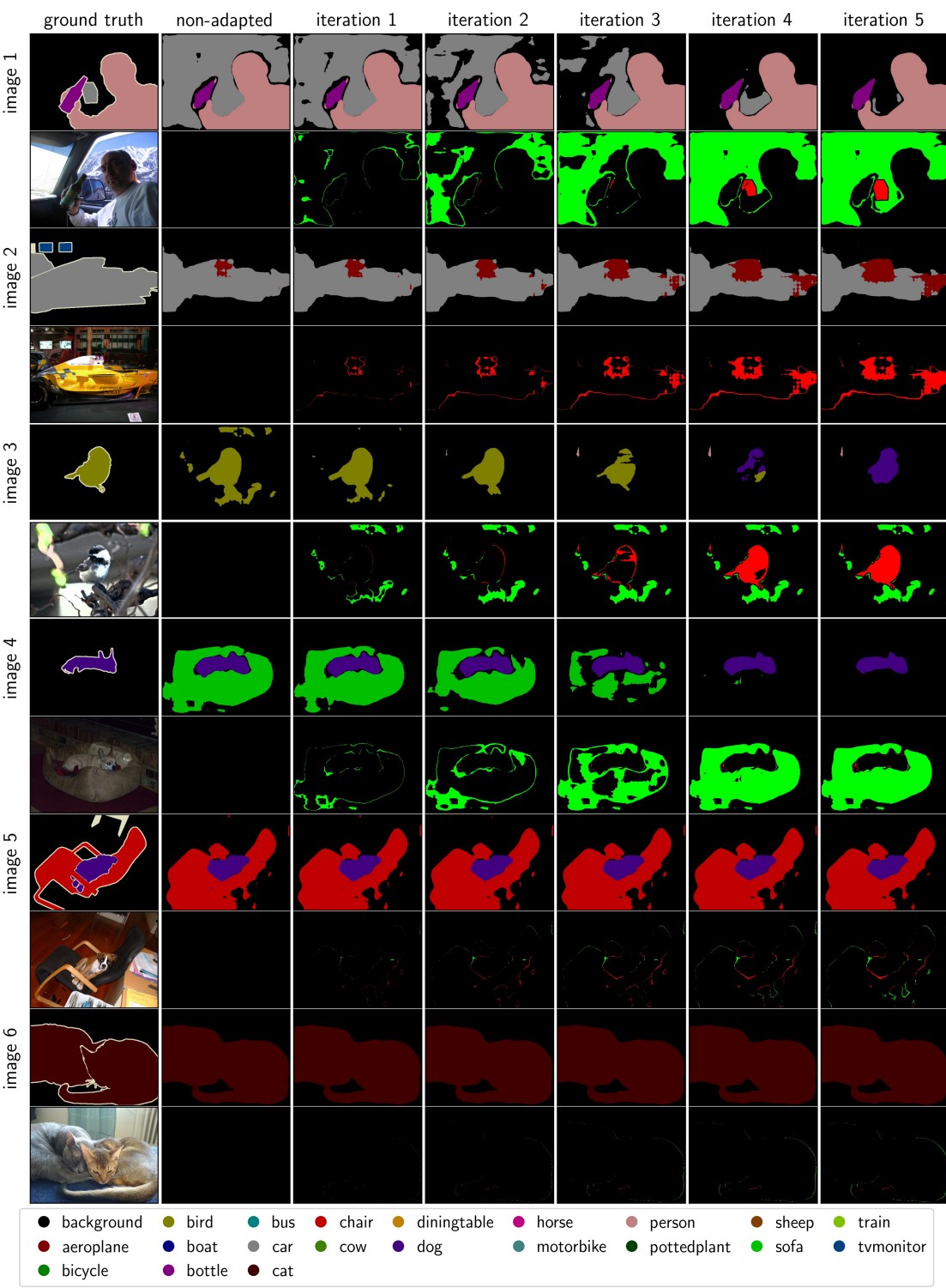

Figure 16: Segmentation evolution during TTA with Ref on VOC test set. First row shows the evolution of masks, second row shows the input image and segmentation improvement w.r.t. to the non-adapted mask. Improved and deteriorated pixels are highlighted.

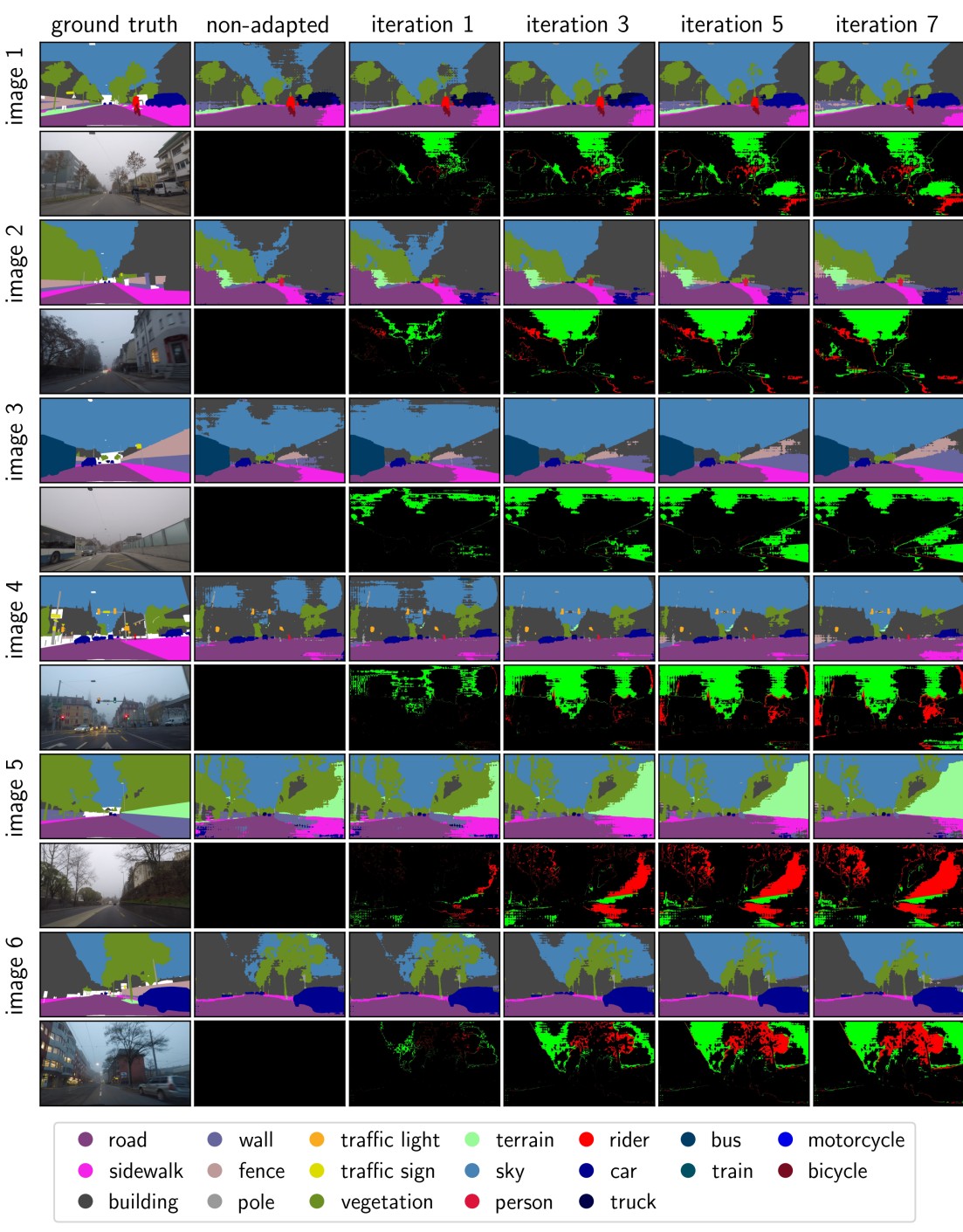

Figure 17: Segmentation evolution during TTA with Ref on ACDC-fog test set. First row shows the evolution of masks, second row shows the input image and segmentation improvement w.r.t. to the non-adapted mask. Improved and deteriorated pixels are highlighted.

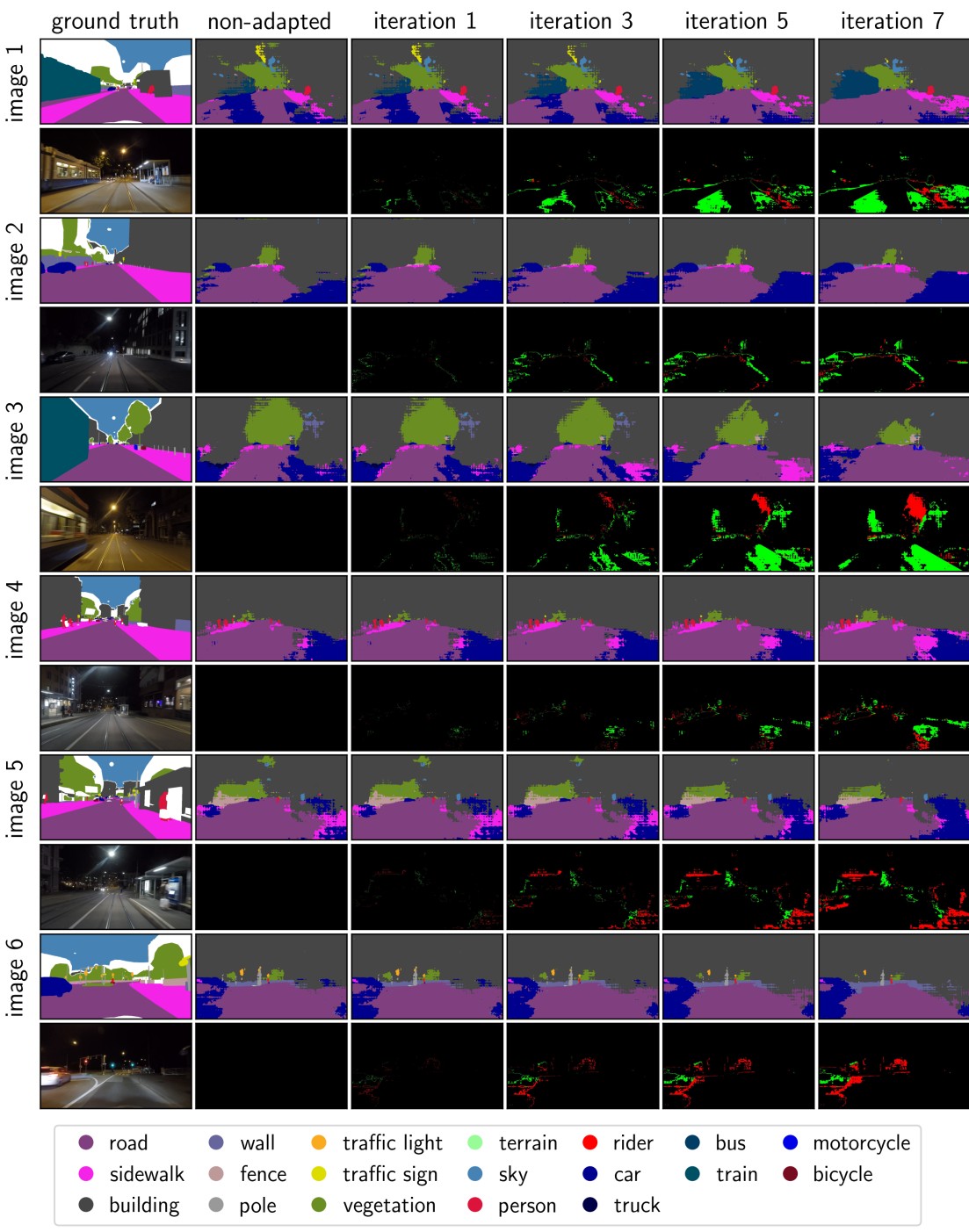

Figure 18: Segmentation evolution during TTA with Ref on ACDC-night test set. First row shows the evolution of masks, second row shows the input image and segmentation improvement w.r.t. to the non-adapted mask. Improved and deteriorated pixels are highlighted.

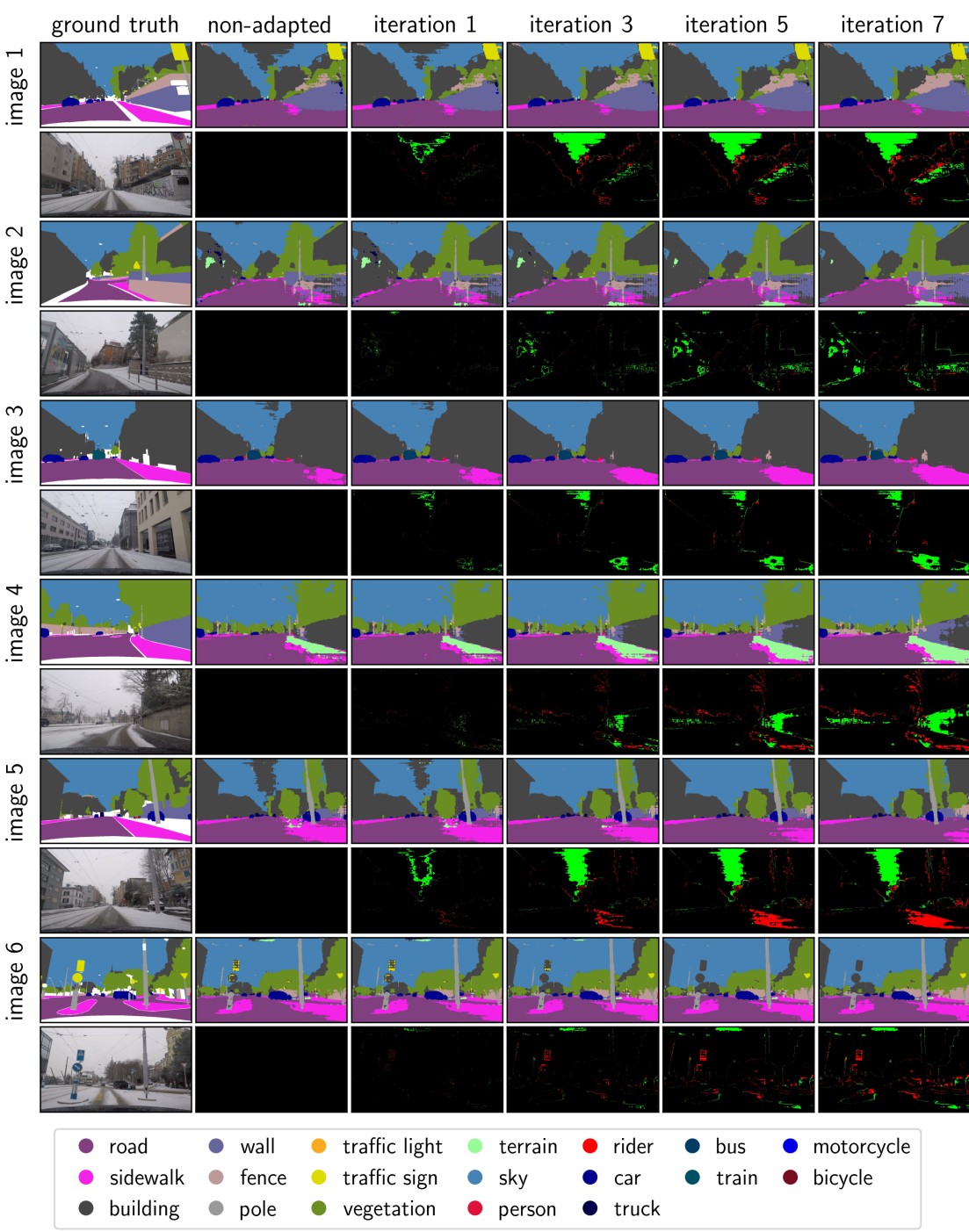

Figure 19: Segmentation evolution during TTA with Ref on ACDC-snow test set. First row shows the evolution of masks, second row shows the input image and segmentation improvement w.r.t. to the non-adapted mask. Improved and deteriorated pixels are highlighted.

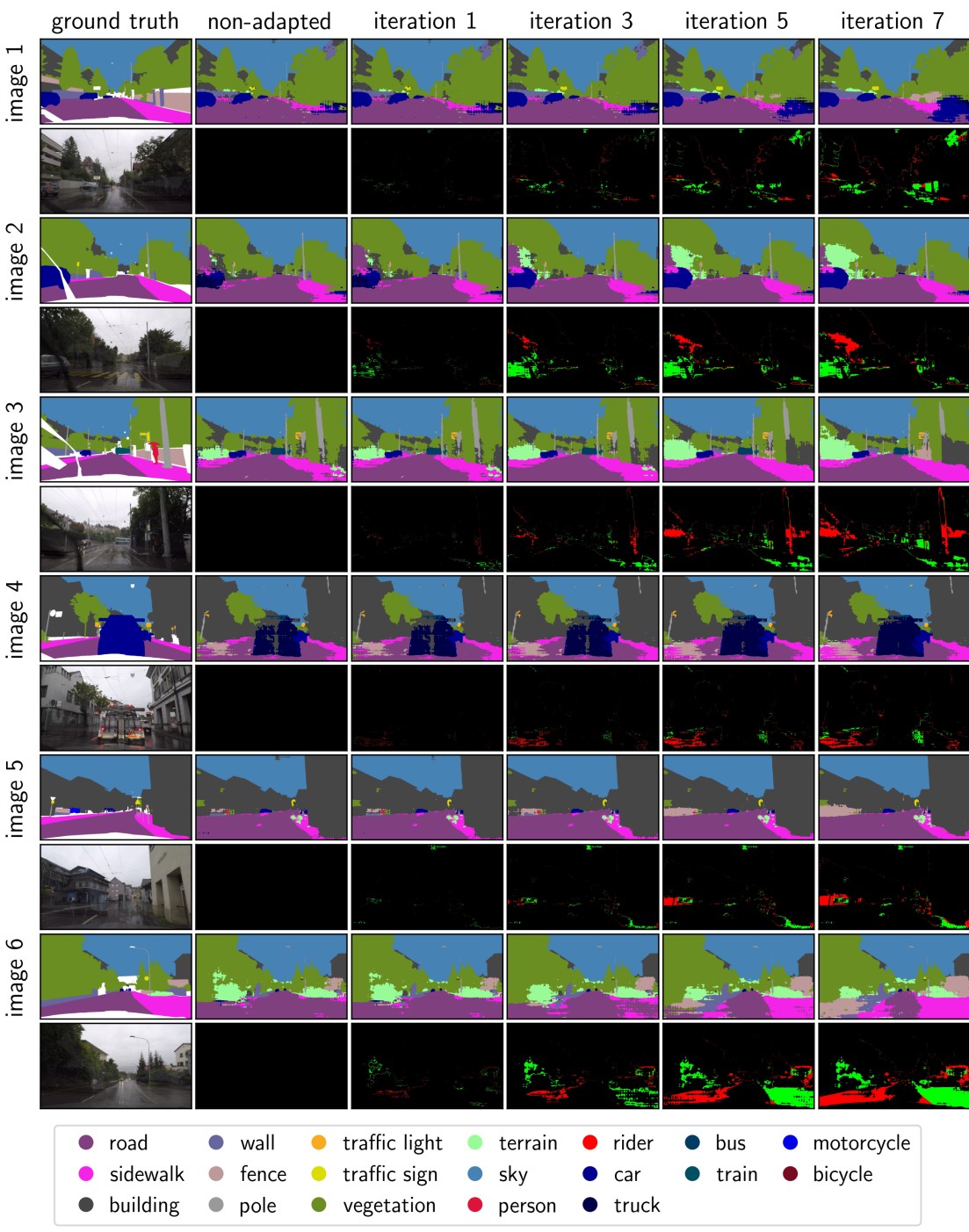

Figure 20: Segmentation evolution during TTA with Ref on ACDC-rain test set. First row shows the evolution of masks, second row shows the input image and segmentation improvement w.r.t. to the non-adapted mask. Improved and deteriorated pixels are highlighted.

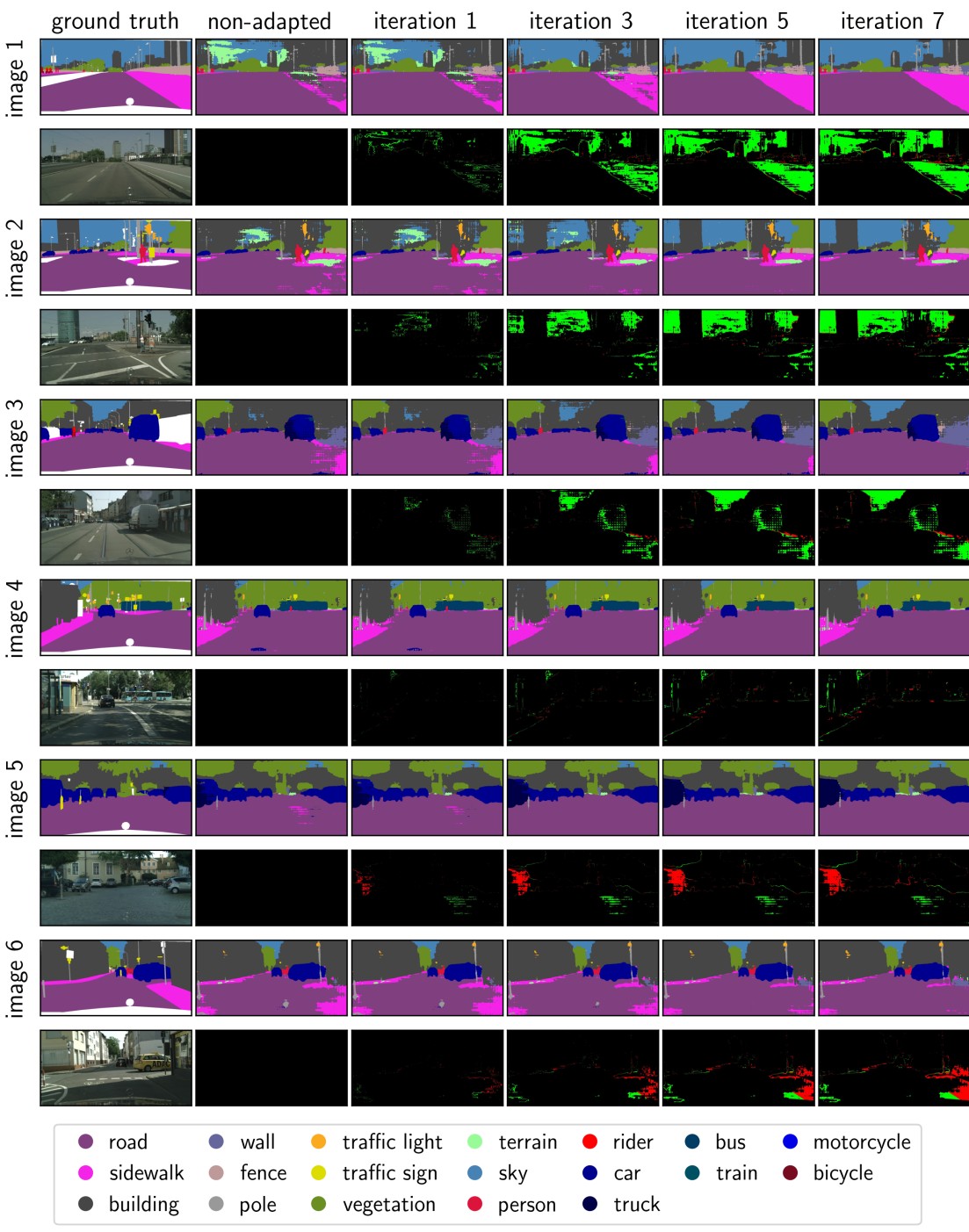

Figure 21: Segmentation evolution during TTA with Ref on cityscapes test set. First row shows the evolution of masks, second row shows the input image and segmentation improvement w.r.t. to the non-adapted mask. Improved and deteriorated pixels are highlighted.