# OpenReview forum: "Single Image Test-Time Adaptation for Segmentation"
_TMLR — Accepted by TMLR_

### Review · Reviewer_dCLn · 2024-02-17

**Summary Of Contributions:**

This paper focuses on test time adaptation of the segmentation task on a single image. The contributions of the paper can be listed as follows:
1. One of the few works studying single image test time adaptation for segmentation.
2. A diverse set of methods are validated on synthetic and real datasets.
3. Replacing Cross-Entropy by IoU loss improves the baseline method performance.

**Audience:**

Yes

**Claims And Evidence:**

Yes

**Requested Changes:**

- More evaluation metrics need to be adopted, such as jaccard index, dice (F1 score), etc.
- Justification of the improvements from replacing the loss, can you provide the intuition on why this is the case?  and more experiments to support this.
- Experiments on more realistic domain shift (with more realistic data)
- Typos: please carefully check the grammar and typos of the submission. e.g. the 4th point of contribution:  "... an underexplroed setup important in applications ...", underexplroed should be underexplored.

**Strengths And Weaknesses:**

Strengths:
- The paper studies an underinvestigated yet important problem -- single image test time adaptation for segmentation.
- The experiments cover a comprehensive set of methods that are considered effective for the problem before.
- The proposed refinement framework, even though not consistently outperform, is simple.

Weakness:
- The proposed method is not performing best in all cases.
- From the experiment results, the proposed SITTA is effective when CE is replaced by IoU, however, no justification for why this is the case.
- regarding the domain shift, the synthetic dataset is generated based on covariate shift, it would be nice to see method performance on datasets a. with label shift b. realistic data.
- only mIOU is used as evaluation metrics are used, but since the IoU is used as the loss function,  it is hard to disentangle the improvements from replacing the training loss as IoU is more directly related to the evaluation metric.

---

> ### Author Response · Authors · 2024-02-25
> **Response to Reviewer dCLn**
>
> We thank the reviewer for the thoughtful feedback.
>
> #### **1.1 From the experiment results, the proposed SITTA is effective when CE is replaced by IoU, however, no justification for why this is the case.**, **1.2. Justification of the improvements from replacing the loss, can you provide the intuition on why this is the case? and more experiments to support this** and **1.3 only mIOU is used as evaluation metrics are used, but since the IoU is used as the loss function, it is hard to disentangle the improvements from replacing the training loss as IoU is more directly related to the evaluation metric**:
> We answer to all these comments jointly as they are closely related.
>
> The mIoU metric has been a predominant evaluation metric for semantic segmentation for a long time. It gives an equal weight to each class, per-class IoU scores are averaged.  Semantic segmentation datasets are known to be highly imbalanced, training with loss functions taking the imbalance into account, such as IoU or Dice loss, is common practice. This is not the case for segmentation TTA methods.
>
> We agree the improvement from using the IoU loss over CE loss is related to its better alignment with the evaluation metric. But we consider this to be a strength rather than a weakness of the paper. The community has agreed to evaluate using this metric, and being able to optimize a metric closer to the evaluation metric is desirable.
>
> We added a discussion of this issue to the experiments section where we discuss the hyper-parameters considered, including the choice of loss function.
>
> #### **2. More evaluation metrics need to be adopted, such as jaccard index, dice (F1 score), etc.**
> We agree it is important to report different evaluation metrics to better understand the performance of different methods. We already compare different IoU-based metrics in Table 4 and 5 in Appendix E. We will move these to the main paper and update them with the Dice score and other metrics in the next revision.
>
> We already report two different metrics based on the Jaccard index (also referred to as IoU), including the standard mIoU, is there another metric based on the Jaccard index we should add?
>
> #### **3. Experiments on more realistic domain shift (with more realistic data)**
> We would like to clarify that while the TTA methods are tuned on synthetic datasets (common practice in domain adaptation literature), all of the test datasets are real-world datasets.
> Label shift is not considered since the methods addressing it are orthogonal and typically discussed in papers focused solely on label shift.
>
> #### **4. Typos: please carefully check the grammar and typos of the submission.**
> We have carefully reviewed the paper with the aid of a spell check and fixed all the typos and grammar issues we found.

---

> ### Comment · Action_Editor_qb9R · 2024-03-28
> **Warning: Please make the final recommmendation immedieately!**
>
> TMLR submission needs your recommendation before we can proceed to the next step.

---

### Review · Reviewer_dkDE · 2024-02-20

**Summary Of Contributions:**

This paper tackles Single Image Test-Time Adaptation for the segmentation task, in which only one image is available for adapting the model. The problem is practically valuable. This paper proposes to use augmentation to alleviate the limited data (only one image) problem and tune the needed hyperparameters in SITTA methods. The authors conducted extensive experiments to analyze and verify the performance of many techniques for tackling this task.

**Audience:**

Yes

**Broader Impact Concerns:**

The authors discuss the broader impact in the main text.

**Claims And Evidence:**

Yes

**Requested Changes:**

My current concern mainly focuses on writing. I will re-evaluate after the re-organization and polishment.

**Strengths And Weaknesses:**

**Strengths**:

1: This paper focuses on a valuable problem that needs to tailor a model for each image in the test-time.

2: The authors conduct extensive experiments to verify the effectiveness of the proposed method.

3: The proposed method seems to be simple yet effective.

**Weaknesses**:

The overall presentation is not ready for publication, and the needed modifications are beyond the scope of the reviewers' responsibility.

1: Grammar and Basic Writing Organization. Here, I take the Introduction as an example, but I believe the problem doesn't only exist in the Introduction.

a) "between source" --> between data sources?

b) "difficult to predict" --> difficult-to-predict?

c) "acsitta not only streamlines": What is "acsitta"? The logical relation between this and the previous sentences is not clear.

d) "the many works relying on the existence ..." --> many works that rely on? Meanwhile, is it possible to convert the batch-normalization to layer-normalization so that such methods can be compatible with transformer architectures?

e) " an additional method from image classification which robustifies the network to small domain shifts (represented by adversarially-attacked images), expecting increased robustness to other domain shifts as well.": The meaning of this sentence is cumbersome to read, and it can be refined.

f) "While medical domain and its domain shifts of interest are different": What does this sentence mean? Why are the domain and domain shifts comparable? I guess the authors want to emphasize the difference between the medical domain and the general non-medical domain.

g) "been on prediction entropy and reconstruction loss" --> "been on predicting entropy and reconstruction loss" or "been on entropy and reconstruction loss prediction"?

h) The summarized contributions are not clear. I can't find what the novel part is, given the simple description, "A novel adversarial refinement module." I want to know the actual improvements rather than a simple sentence, "improvements of baselines."

2: Format.

a) Tables (both Table 1 and Table 2) are beyond the border of the main paper.

b) Legends and main content overlap are not encouraged (Figure 6). Why don't authors align subfigures together?

c) In Figure 8, the left and right subfigures are not aligned well. The text and number on the x-axis overlap.

3: Emphasizing the Proposed Technique Contribution.

From the method section and Figure 2, I can't see what the proposed technique contribution is. Figure 2 presents the pipeline of Mask Refinement, but the idea has been proposed in medical imaging, and the adversarial-attack is also proposed before. The only contribution seems to test their effectiveness in the non-medical domain. Meanwhile, the shown inference and training images are the same, which doesn't suggest the challenge of test-time adaptation i.e., the domain gap between training and test images.

All in all, I appreciate the authors' efforts in conducting extensive experiments on the single image test-time adaptation tasks. However, it's hard for me to evaluate the contribution of this paper, given the current presentation. I feel this paper was written in a hurry and kindly encourage authors to re-organize and polish the paper, especially in emphasizing the paper's contribution. I look forward to the revised version and will re-evaluate this paper.

---

> ### Author Response · Authors · 2024-02-25
> **Response to Reviewer dkDE**
>
> We thank the reviewer for the thorough feedback. We have submitted a revision addressing all the issues. In particular, the introduction, conclusions and methods section is updated to better emphasize our contributions and differentiate our method from prior work.
>
> Please let us know if there are further concerns.

---

> > ### Comment · Reviewer_dkDE · 2024-03-02
> >
> > Thanks for the authors' revised paper. I have carefully read the main text and supplementary material.
> >
> > 1: The overall writing improves much for this version, although I still find some minor typos and sentences that need to be refined.
> >
> > a) "Single Image Test-Time Adaptation (SITTA) tailors a **segmentation model**" on Page 1. I think the SITTA is not specific to the segmentation task, and the overall paragraph is also not concentrated on the segmentation task.
> >
> > b) "tzeng2017adversarial" on Page 2 is not applied cross-reference.
> >
> > c) " tta:! (tta:!)" on Page 8. I don't know what this means.
> >
> > The authors are encouraged to check potential typos since there may be other typos that I missed.
> >
> > 2: For the method and experiments part, this paper conducts comprehensive experiments to verify the key point for the SITTA task, including four baseline methods and two newly proposed (modified) methods. Although the proposed methods may sound simple, I believe they are effective and can be strong baselines for the community to explore SITTA's key challenges further.
> >
> > I'm leaning toward accepting this paper, given the current comprehensive baseline results and the importance of the SITTA task. I also encourage the authors to open-source their codes since the main contributions of this paper mainly focus on baseline methods with reproducible results and evaluation metrics to measure the effectiveness of the SITTA methods. I'm open to discussions with AE and other reviewers.
> >
> > Meanwhile, I'm willing to opt for “Reproducibility Certification” if the authors could provide the code.

---

> > > ### Author Response · Authors · 2024-03-02
> > > **Response to Reviewer dkDE**
> > >
> > > Thank you for re-evaluating the paper. We will carefully incorporate the new feedback into the second revision, and will again check for any leftover typos/....
> > >
> > > We do plan to open-source the code.

---

### Review · Reviewer_cWBQ · 2024-02-21

**Summary Of Contributions:**

This paper focuses on a novel setting, namely single image test-time adaptation. Evaluation metric and framework are developed, and a novel method is proposed to tackle the proposed task.

**Audience:**

Yes

**Broader Impact Concerns:**

No broader impact discussion are needed in the reviewer's opinion.

**Claims And Evidence:**

Yes

**Requested Changes:**

No requested changes.

**Strengths And Weaknesses:**

Strength:
1. The proposed task is interesting and novel.
2. Complete framework including evaluation metrics, baseline, etc, are developed for this new task.
3. A novel and effective method is proposed to tackle the proposed setting.
4. Thorough empirical study are provided to validate the effectiveness of the method.

Weakness:
No prominent weakness.

---

> ### Comment · Action_Editor_qb9R · 2024-03-28
> **Warning: Please make the final recommmendation immedieately!**
>
> TMLR submission needs your recommendation before we can proceed to the next step.

---

### Author Response · Authors · 2024-02-25
**To All Reviewers**

We thank all the reviewers for their feedback.

We have submitted a first revision with a significantly rewritten introduction and conclusions to better emphasize the contributions and improve clarity, as requested by Reviewer dkDE. We checked the writing with a spell check and had the contents reviewed by colleagues.

We are working on a second revision focused on additional evaluation metrics and further improving any unclear parts.

---

### Comment · Action_Editor_qb9R · 2024-02-26
**To All Reviewers: Please Re-evaluate This Paper and Provide a Final Decision Below Your Review**

The author has revised the paper based on the comments of the reviewers. Please review the latest version and make a final decision。

---

> ### Author Response · Authors · 2024-02-26
> **Clarification regarding revision**
>
> We would like to clarify we updated the revised paper based on Reviewer-dkDE's feedback to see if there are further concerns we should address and we are still working on the additional evaluation, as requested by Reviewer-dCLn.

---

### Author Response · Authors · 2024-03-06
**To All Reviewers: Revision Part II**

We thank the reviewers for their suggestions on improving the paper.

We have uploaded a second revision of the paper focused on additional evaluation metrics, as requested by Reviwer dCLn.

1. Tables 2 and 4 with real-world test dataset results were moved to the main paper (previously in Appendix E). These tables were extended with multiple new metrics (mIoU, mDice, total accuracy)
2. Fixed typos/grammar issues (including those pointed out by Reviewer dkDE)
5. Minor text revisions to improve clarity in the experiments section.

Our top priority now is to polish the code to open-source it as soon as possible.

---

### Comment · Action_Editor_qb9R · 2024-03-20
**To All Reviewers: Please Make the Final Recommendation**

Please make the final recommendation, the deadline is 21 Mar. 2024

---

### Decision · Action_Editor_qb9R · 2024-04-04

**Recommendation:** Accept with minor revision

**Comment:**

All the reviewers lean to acceptance after the rebuttal. However, the authors still need to address the minor remaining issues. Specifically, the reviewers suggested releasing codes or relative materials, which makes this work reproducible. Moreover, the authors should further proofread the manuscript and fix all the typos.

**Audience:**

The audience, particularly who focus on domain adaptation or segmentation, will be interested in this paper.

**Claims And Evidence:**

This work explores propose a segmentation model which dose not use data available at test-time. The claims of this work are made based on this setting and supported by sufficient experimental results.